

# Anti-clustering in the national SARS-CoV-2 daily infection counts

Boudewijn F. Roukema[1,2]

[1] Institute of Astronomy, Faculty of Physics, Astronomy and Informatics, ul. Grudziadzka 5, Nicolaus Copernicus University of Torun, Torun, Poland
[2] Univ Lyon, Ens de Lyon, Univ Lyon1, CNRS, UMR5574, Centre de Recherche Astrophysique de Lyon, Lyon, France

Corresponding author
Boudewijn F. Roukema,
boud@astro.uni.torun.pl

## ABSTRACT

The noise in daily infection counts of an epidemic should be super-Poissonian due to intrinsic epidemiological and administrative clustering. Here, we use this clustering to classify the official national SARS-CoV-2 daily infection counts and check for infection counts that are unusually anti-clustered. We adopt a one-parameter model of $\phi_i'$ infections per cluster, dividing any daily count $n_i$ into $n_i/\phi_i'$ 'clusters', for 'country' $i$. We assume that $n_i/\phi_i'$ on a given day $j$ is drawn from a Poisson distribution whose mean is robustly estimated from the four neighbouring days, and calculate the inferred Poisson probability $P_{ij}'$ of the observation. The $P_{ij}'$ values should be uniformly distributed. We find the value $\phi_i$ that minimises the Kolmogorov–Smirnov distance from a uniform distribution. We investigate the $(\phi_i, N_i)$ distribution, for total infection count $N_i$. We consider consecutive count sequences above a threshold of 50 daily infections. We find that most of the daily infection count sequences are inconsistent with a Poissonian model. Most are found to be consistent with the $\phi_i$ model. The 28-, 14- and 7-day least noisy sequences for several countries are best modelled as sub-Poissonian, suggesting a distinct epidemiological family. The 28-day least noisy sequence of Algeria has a preferred model that is strongly sub-Poissonian, with $\phi_i^{28} < 0.1$. Tajikistan, Turkey, Russia, Belarus, Albania, United Arab Emirates and Nicaragua have preferred models that are also sub-Poissonian, with $\phi_i^{28} < 0.5$. A statistically significant ($P^\tau < 0.05$) correlation was found between the lack of media freedom in a country, as represented by a high Reporters sans frontieres Press Freedom Index (PFI[2020]), and the lack of statistical noise in the country's daily counts. The $\phi_i$ model appears to be an effective detector of suspiciously low statistical noise in the national SARS-CoV-2 daily infection counts.

## INTRODUCTION

The daily counts of new, laboratory-confirmed infections with severe acute respiratory syndrome coronavirus 2 (SARS-CoV-2) constitute one of the key statistics followed by citizens and health agencies around the world in the ongoing 2019–2020 Coronavirus Disease 2019 (COVID-19) pandemic (*Huang et al., 2020a*; *Li et al., 2020*). Can these counts be classified in a way that makes as few epidemiological assumptions as possible, as

motivation for deeper analysis to either validate or invalidate the counts? While full epidemiological modelling and prediction is a vital component of COVID-19 research (*Chowdhury et al., 2020*; *Kim et al., 2020*; *Molina-Cuevas, 2020*; *Jiang, Zhao & Shao, 2021*; *Afshordi et al., 2020*), these cannot be accurately used to study the pandemic as a whole—a global phenomenon by definition—if the data at the global level is itself inaccurate. Knowledge of the global state of the current pandemic is weakened if any of the national-level SARS-CoV-2 infection data have been artificially interfered with by the health agencies providing that data or by other actors involved in the chain of data lineage (*Thomas et al., 2017*). Since personal medical data are private information, only a limited number of individuals at health agencies are expected to be able to check the validity of these counts based on original records. Nevertheless, artificial interventions in the counts could potentially reveal themselves in statistical properties of the counts. Unusual statistical properties in a wide variety of quantitative data sometimes appear, for example, as anomalies related to Benford's law (*Newcomb, 1881*; *Nigrini & Miller, 2009*), as in the 2009 first round of the Iranian presidential election (*Roukema, 2014*, *2015*; *Mebane, 2010*). Benford's law analysis has been used to argue that countries with higher democracy indices, high gross domestic product, and better health system indices tend to have a lower probability of having manipulated their key COVID-19 related cumulative counts (confirmed cases and deaths, *Balashov, Yan & Zhu, 2021*). For other Benford's law COVID-19 count analyses, see *Koch & Okamura (2020)* and *Lee, Han & Jeong (2020)*. For a case-specific analysis of the lack of noise in governmental medical data, see the analysis of official deceased-donor organ-donation data from China (*Robertson, Hinde & Lavee, 2019*), using methods different to the one introduced in this article. For the politics of organisational strategies regarding open government data, see *Ruijer et al. (2019)*.

Here, we check the compatibility of noise in the official national SARS-CoV-2 daily infection counts, $n_i(t)$, for country[1] $i$ on date $t$, with expectations based on the Poisson distribution (*Poisson (1837)*; for a review, see, *e.g.*, *Johnson, Kemp & Kotz, 2005*). The Poisson distribution is motivated by the one-day time scale for an infection count being several times shorter than the dominant time scale involved, the incubation time scale, estimated at about five days (*Lauer et al., 2020*; *Yang et al., 2020*), with a 95% confidence interval (CI) from about one to 15 days (*Yang et al., 2020*). Since each infected person typically infects about two to three others ($R_0 \sim 2.4$–$3.3$ at 95% (CI), *Billah, Miah & Khan, 2020*), these secondarily infected people would typically be assessed as SARS-CoV-2 positive on independent days, if they were diagnosed immediately after the onset of symptoms, with instantaneous laboratory testing and test results reported instantly in the official national count data. In reality, delays for diagnosis, testing and reporting and collating the test results are random processes which should further add delays that reduce correlations among positive test results between distinct nearby days; a Poissonian process is a simple hypothesis for each of these separate processes. Poisson processes are both additive and infinitely divisible (*Johnson, Kemp & Kotz, 2005*, "Discussion"), so the combination of these processes can reasonably yield an overall Poisson process.

[1] No position is taken in this paper regarding jurisdiction over territories; the term "country" is intended here as a neutral term without supporting or opposing the formal notion of state. Apart from minor changes for technical reasons, the countries are defined by the data sources.

However, it is unlikely that any real count data will be fully modelled by a Poisson distribution, both due to the complexity of the logical tree of time-dependent intrinsic epidemiological infection as well as administrative effects in the SARS-CoV-2 testing procedures, and the sub-national and national level procedures for collecting and validating data to produce a national health agency's official report. In particular, clusters of infections on a scale of $\phi_i'$ infections per cluster, either intrinsic or in the testing and administrative pipeline, would tend to cause relative noise to increase from a fraction of $1/\sqrt{n_i}$ for pure Poisson noise up to $\sqrt{\phi_i'/n_i}$, greater by a factor of $\sqrt{\phi_i'}$. This overdispersion has been found, for example, for SARS-CoV-2 transmission (*Endo et al., 2020*; *He et al., 2020*) and for COVID-19 death rate counts in the United States (*Kim et al., 2020*).

In contrast, it is difficult to see how anti-Poissonian smoothing effects could occur, unless they were imposed administratively. For example, an administrative office might impose (or have imposed on it by political authorities) a constraint to validate a fixed or slowly and smoothly varying number of SARS-CoV-2 test result files per day, independently of the number received or queued; this would constitute an example of an artificial intervention in the counts that would weaken the epidemiological usefulness of the data.

A one-parameter model to allow for the clustering is proposed in this paper, and used to classify the counts. We allow the parameter to take on an effective anti-clustering value, in order to allow the data to freely determine its optimal value, without forcing overdispersion. While a distribution of clustering values for a given country is likely to be more realistic than a single value, Occam's razor favours adding as few parameters as possible. For example, a power-law distribution of arbitrary (negative) index would require a second parameter to truncate the tail in non-convergent cases. While the one-parameter anti-clustering value is a simplified model, it has the advantage of allowing a straightforward, though simplified, interpretation in terms of clustering. If the one-parameter method proposed here is found to viable, then the method could be extended by including models of directly observed estimates of SARS-CoV-2 clustering.

As an alternative to this clustering model, we also consider a negative binomial distribution (*e.g. Johnson, Kemp & Kotz, 2005*, "Conclusion"). *Lloyd-Smith et al. (2005)* found the negative binomial distribution, as a mix of Poisson distributions over a Gamma distribution, to be better at modelling secondary infections by SARS-CoV-1 (and other infectious agents) than Poisson and geometric distributions, quantifying what are referred to as superspreader events in an epidemic. This has also been found to be relevant to SARS-CoV-2 secondary infections (*Endo et al., 2020*; *He et al., 2020*). However, since the negative binomial model only allows overdispersion with respect to the Poisson model, it is unlikely to provide the best model for data which may have been artificially modified to the extent of becoming sub-Poissonian. More in-depth models of clustering, called burstiness in stochastic models of discrete event counts, include power-law models (*Barabási, 2005*; *Goh & Barabasi, 2006*).

The method is presented in "Method". "SARS-CoV-2 Infection Data" describes the choice of data set and the definition, for any given country, of a consecutive time sequence

that has high enough daily infection counts for Poisson distribution analysis to be reasonable. The method of analysis is given in "Primary Analysis" for full sequences ("Poissonian and $\phi_i'$ Models: Full Sequences"), subsequences ("Subsequences") and alternatives to the main method ("Alternative Analyses"). Results are presented in "Results". A non-parametric comparison with the *Reporters sans frontières* Press Freedom Index, which should not have any relation to noise in SARS-CoV-2 daily counts in the absence of a sociological connection, is provided in "Comparison with the RSF Press Freedom Index". Qualitative discussion of the results is given in "Discussion". Conclusions are summarised in "Conclusion". This work is intended to be fully reproducible by independent researchers using the Maneage framework; it was produced using commit f72cb84 of the live Git repository https://codeberg.org/boud/subpoisson on a computer with Little Endian x86_64 architecture; the source is archived at zenodo.4765705 and swh:1:rev:789e651c0fb23b2585555c08de1b44d9e25cfb6d.

## METHOD

### SARS-CoV-2 infection data

Two obvious choices of a dataset for national daily SARS-CoV-2 counts would be those provided by the World Health Organization (WHO), (archive: https://covid19.who.int/WHO-COVID-19-global-data.csv) or those curated by the Wikipedia *WikiProject COVID-19 Case Count Task Force* (https://en.wikipedia.org/w/index.php?title=Wikipedia:WikiProject_COVID-19/Case_Count_Task_Force&oldid=1001119689) in *medical cases chart* templates (hereafter, C19CCTF ). While WHO has published a wide variety of documents related to the COVID-19 pandemic, it does not appear to have published details of how national reports are communicated to it and collated. Given that most government agencies and systems of government procedures tend to lack transparency, despite significant moves towards forms of open government (*Yu & Robinson, 2012*) in many countries, data lineage tracing from national governments to WHO is likely to be difficult in many cases. In contrast, the curation of official government SARS-CoV-2 daily counts by the Wikipedia *WikiProject COVID-19 Case Count Task Force* follows a well-established technology of tracking data lineage. The Wikipedia community high-tempo collaborative editing that has taken place in response to the COVID-19 pandemic is well quantified (*Keegan & Tan, 2020*). The John Hopkins University Center for Systems Science and Engineering curated set of official COVID-19 data is discussed below.

Unfortunately, it is clear that in the WHO data, there are several cases where two days' worth of detected infections appear to be listed by WHO as a sequence of two days $j$ and $j + 1$ on which all the infections are allocated to the second of the two days, with zero infections on the first of the pair. There are also some sequences in the WHO data where the day listed with zero infections is separated by several days from a nearby day with double the usual amount of infections. This is very likely an effect of difficulties in correctly managing world time zones, or time zone and sleep schedule effects, in any of several levels of the chains of communication between health agencies and WHO. In other words, there are several cases where a temporary sharp jump or drop in the counts

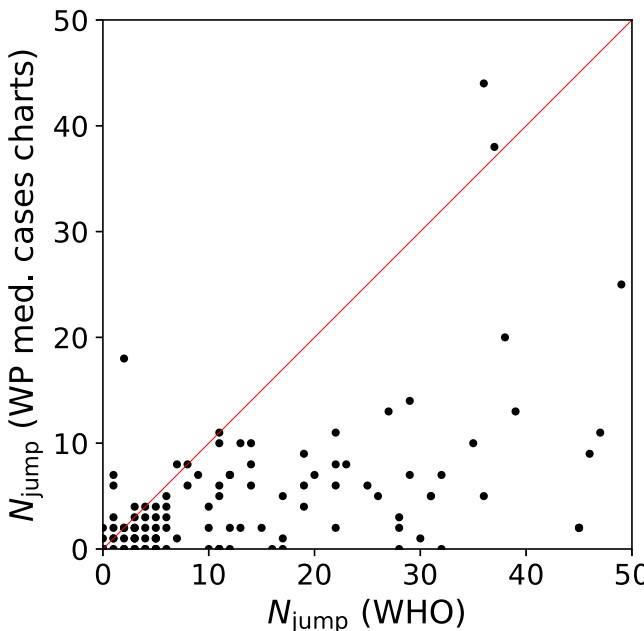

**Figure 1 Number $N_{jump}$ of sudden jumps or drops in counts on adjacent days in WHO and Wikipedia *WikiProject COVID-19 Case Count Task Force medical cases chart* national daily SARS-CoV-2infection counts for countries present in both data sets.** A line illustrates equal quality of the two datasets. The C19CCTF version of the data is clearly less affected by sudden jumps than the WHO data. Plain-text table: zenodo.4765705/WHO_vs_WP_jumps.dat.

appears in the data but is reasonably interpreted as a timing artefact. Whatever the reason for the effect, this effect will tend to confuse the epidemiological question of interest here: the aim is to globally characterise the noise and to highlight countries where unusual smoothing may have taken place.

We quantify this jump/drop problem as follows. We consider a pair of days $j, j + 1$ for a given country to be a jump if the absolute difference in counts, $|n_i(j + 1) - n_i(j)|$, is greater than the mean, $(n_i(j + 1) + n_i(j))/2$. In the case of a pair in which one value is zero, the absolute difference is twice the mean, and the condition is necessarily satisfied. We evaluate the number of jumps $N_{jump}$ for both the WHO data and the C19CCTF *medical cases chart* data, starting, for any given country, from the first day with at least 50 infections. Figure 1 shows $N_{jump}$ for the 137 countries in common to the two data sets; there are 237 countries in the WHO data set and 139 in the C19CCTF data. It is clear that most countries have fewer jumps or drops in the Wikipedia data set than in the WHO data set.

Thus, at least for the purposes of understanding intrinsic and administrative clustering, the C19CCTF *medical cases chart* data appear to be the better curated version of the national daily SARS-CoV-2 infection counts as reported by official agencies. The detailed download and extraction script of national daily SARS-CoV-2 infection data from these templates and the resulting data file zenodo.4765705/WP_C19CCTF_SARSCoV2. dat (downloaded 6 May 2021) are available in the reproducibility package associated with

this paper (§Code availability). Dates without data are omitted; this should have an insignificant effect on the analysis if these are due to low infection counts.

Another global collection of daily SARS-CoV-2 counts that could be considered is the John Hopkins University Center for Systems Science and Engineering (JHU CSSE) git repository. Unfortunately, for several countries, the JHU CSSE data are provided for sub-national divisions rather than as official national statistics, making the dataset inhomogeneous for the purposes of this study. Artificial interference in the data at the national level will not be shown in data that is the sum of data obtained directly from sub-national geographical/political divisions. Moreover, detailed data provenance analysis (Which exact government URL did a particular count come from? Where is the archived version of the data of the original URL?) appears to be more difficult for the JHU CSSE data than for the C19CCTF data. Nevertheless, for completeness, the JHU CSSE data is analysed using the same method as the main analysis, with results presented as tables in Appendix A.

The full set of C19CCTF data includes many days, especially for countries or territories (as defined by the data source) of low populations, with low values, including zero and one. The standard deviation of a Poisson distribution (*Poisson, 1837*) of expectation value $N$ is $\sqrt{N}$, giving a fractional error of $1/\sqrt{N}$. Even taking into account clustering or anticlustering of data, inclusion of these periods of close to zero infection counts would contribute noise that would overwhelm the signal from the periods of higher infection rates for the same or other countries. In the time sequences of SARS-CoV-2 infection counts, chaos in the administrative reactions to the initial stages of the pandemic will tend to create extra noise, so it is reasonable to choose a moderately high threshold at which the start and end of a consecutive sequence of days should be defined for analysis. Here, we set the threshold for a sequence to start as a minimum of 50 infections in a single day. The sequence is continued for at least 7 days (if available in the data), and stops when the counts drop below the same threshold for 2 consecutive days. The cutoff criterion of 2 consecutive days avoids letting the analysable sequence be too sensitive to individual days of low fluctuations. If the resulting sequence includes less than 7 days, the sequence is rejected as having insufficient signal to be analysed.

## RSF press freedom index

The *Reporters sans frontières* (RSF) Press Freedom Index is derived annually from an 87-question survey translated into 20 languages and sent to media professionals, lawyers and sociologists from 180 countries, yielding scores on six general criteria of media freedom and a weighted score representing executions, imprisonments, kidnappings and related abuses against journalists (*Reporters sans frontieres, 2021*). The scores are combined into an overall score from zero (best) to 100 (worst) that we denote here as $\mathrm{PFI}^{2020}$.

In the absence of artificial interference in the SARS-CoV-2 daily counts, there is no obvious reason why media freedom should relate to the noise in the SARS-CoV-2 counts. However, a correlation between the lack of media freedom and the publication of manipulated data by government agencies would not be surprising. Governments and the public service as organisations, and the individuals that compose them, are under more

[2] (https://rsf.org/en/ranking/2020, downloaded 4 May 2021)

pressure to be honest in places and epochs where there is more press freedom. To see if the hypothesis of artificial interference is credible, the results of the current work are compared with $PFI^{2020}$, as published for 2020[2], in "Comparison with the RSF Press Freedom Index".

## Primary analysis

### Poissonian and $\phi_i'$ models: full sequences

We first consider the full count sequence $\{n_i(j), 1 \leq j \leq T_i\}$ for each country $i$, with $T_i$ valid days of analysis as defined in "SARS-CoV-2 Infection Data". Our one-parameter model assumes that the counts are predominantly grouped in clusters, each with $\phi_i'$ infections per cluster. Thus, the daily count $n_i(j)$ is assumed to consist of $n_i(j)/\phi_i'$ infection events. We assume that $n_i(j)/\phi_i'$ on a given day is drawn from a Poisson distribution of mean $\hat{\mu}_i(j)/\phi_i'$. We set $\hat{\mu}_i(j)$ to the median of the 4 neighbouring days, excluding day $j$ and centred on it. For the initial sequence of 2 days, $\hat{\mu}_i(j)$ is set to $\hat{\mu}_i(3)$, and $\hat{\mu}_i(j)$ for the final 2 days is set to $\hat{\mu}_i(T_i - 2)$. By modelling $\hat{\mu}_i$ as a median of a small number of neighbouring days, our model is almost identical to the data itself and statistically robust, with only mild dependence on the choices of parameters. This definition of a model is more likely to bias the resulting analysis towards underestimating the noise on scales of several days rather than overestimating it; this method will not detect oscillations on the time scale of a few days to a fortnight that are related to the SARS-CoV-2 incubation time (*Lauer et al., 2020*; *Yang et al., 2020*; *Huang et al., 2020b*). For any given value $\phi_i'$, we calculate the cumulative probability $P_{ij}'$ that $n_i(j)/\phi_i'$ is drawn from a Poisson distribution of mean $\hat{\mu}_i(j)/\phi_i'$. For country $i$, the values $P_{ij}'$ should be drawn from a uniform distribution if the model is a fair approximation. In particular, for $\phi_i'$ set to unity, $P_{ij}'$ should be drawn from a uniform distribution if the intrisic data distribution is Poissonian. Individual values of $P_{ij}'$ (those that are close to zero or one) could, in principle, be used to identify individual days that are unusual, but here we do not consider these further.

We allow a wide logarithmic range in values of $\phi_i'$, allowing the unrealistic subrange of $\phi_i' < 1$, and find the value $\phi_i$ that minimises the Kolmogorov–Smirnov (KS) distance (*Kolmogorov, 1933*; *Smirnov, 1948*; *Justel, Pen & Zamar, 1997*; *Marsaglia, Tsang & Wang, 2003*) from a uniform distribution, *i.e.* that maximises the KS probability that the data are consistent with a uniform distribution, when varying $\phi_i'$. The one-sample KS test is a non-parametric test that compares a data sample with a chosen theoretical probability distribution, yielding the probability that the sample is drawn randomly from the theoretical distribution. This test uses information from the whole of the reconstructed cumulative distribution function, *i.e.* the set of $P_{ij}'$ values for a given country $i$. We label the corresponding KS probability as $P_i^{KS}$. We write $P_i^{POISS} := P_i^{KS}(\phi_i' = 1)$ to check if any country's daily infection rate sequence is consistent with Poissonian, although this is likely to be rare, as stated above: super-Poissonian behaviour seems reasonable. Of particular interest are countries with low values of $\phi_i$. Allowing for a possibly fractal or other power-law nature of the clustering of SARS-CoV-2 infection counts, we consider the possibility that the optimal values $\phi_i$ may be dependent on the total infection count $N_i$. We investigate the $(\phi_i, N_i)$ distribution and see whether a scaling type relation exists,

allowing for a corrected statistic $\psi_i$ to be defined in order to highlight the noise structure of the counts independent of the overall scale $N_i$ of the counts.

Standard errors in $\phi_i$ for a given country $i$ are estimated once $\phi_i$ has been obtained by assuming that $\hat{\mu}_i(j)$ and $\phi_i$ are correct and generating *30* Poisson random simulations of the full sequence for that country. Since the scales of interest vary logarithmically, the standard deviation of the best estimates of $\log_{10} \phi_i$ for these numerical simulations is used as an estimate of $\sigma(\log_{10}\phi_i)$, the logarithmic standard error in $\phi_i$.

### Subsequences

Since artificial interference in daily SARS-CoV-2 infection counts for a given country might be restricted to shorter periods than the full data sequence, we also analyse 28-, 14- and 7-day subsequences. These analyses are performed using the same methods as above ("Poissonian and $\phi_i'$ Models: Full Sequences"), except that the 28-, 14- or 7-day subsequence that minimises $\phi_i$ is found. The search over all possible subsequences would require calculation of a Šidàk–Bonferonni correction factor (*Abdi, 2007*) to judge how anomalous they are. The KS probabilities that we calculate need to be interpreted keeping this in mind. Since the subsequences for a given country overlap, they are clearly not independent from one another. Instead, the *a posteriori* interpretation of the results of the subsequence searches found here should at best be considered indicative of periods that should be considered interesting for further verification.

## Alternative analyses

Alternatives to the method presented in "Poissonian and $\phi_i'$ Models: Full Sequences" are checked to see if they provide better models of the data.

### Logarithmic median model

Each country's time series is by default modelled with the mean of the expected Poisson distribution for $n_i(j)/\phi_i'$ on a given day being $\hat{\mu}_i(j)/\phi_i'$, where $\hat{\mu}_i(j)$ is the median of $n_i$ in the 4 neighbouring days, excluding day $j$ and centred on it. As an alternative, we replace $\hat{\mu}_i(j)$ on day $j$ by $v_i(j) := \exp(\text{median}(\ln(n_i)))$ calculated over the same set of neighbouring days. That is, we replace the usual linear median by a logarithmic median. This might better model the growing and decaying exponential phases of the infection count sequence.

### Negative binomial model

The negative binomial distribution forbids underdispersion, but is worth considering, given its epidemiological motivation for the step from primary to secondary infections (*Lloyd-Smith et al., 2005*; *Endo et al., 2020*; *He et al., 2020*). For the counts of a given country $i$, we define an overdispersion parameter $\omega_i'$, where the binomial probability mass function for a given infection count $k$, considered as $k$ failures, compared to $r$ successes, with a probability $p$ of success, is

$$P(k; n, p) = \binom{k + r - 1}{k}(1 - p)^k p^r \tag{1}$$

$$p := \frac{\omega'_i}{1 + \omega'_i}. \tag{2}$$

On day $j$, with a modelled count of $_i(j)$, we set

$$r := \omega'_i \widehat{\mu}_i(j), \tag{3}$$

giving $\hat{\mu}_i(j)$ as the mean of the distribution and $\hat{\mu}_i(j)\,(1 + \omega'_i)$ as the variance. The preferred value of $\omega'_i$ (that yielding the lowest Kolmogorov–Smirnov test statistic when comparing the set of cumulative probabilities with a uniform distribution, as in "Poissonian and $\phi'_i$ Models: Full Sequences") is then $\omega_i$. Thus, $\omega_i$ should behave similarly to $\phi_i$ to represent typical cluster size when both are greater than unity, while at low values (below unity), $\omega_i$ will be unable to represent distributions that are underdispersed with respect to the Poisson distribution, and will instead rapidly approach zero (the Poisson limit).

### *Does anti-clustering exist in grouped data?*

The temptation to make 'unnoticeable' modifications that hide an increase in data from day $j$ to day $j + 1$ might be less likely to occur on greater timescales. Moreover, some of the phenomena contributing to the intrinsic and administrative components of $\phi'_i$ should be independent of time scale, while others should depend on the time scale. To provide clues for this type of analysis, the $n_i(j)$ data have been summed in pairs and triplets of days, ignoring any one- or two-day remainder at the end of a sequence. These were analysed using the same algorithm as above for the full sequences ("Poissonian and $\phi'_i$ Models: Full Sequences").

### *Akaike and Bayesian information criteria*

In each case we calculate the *Akaike (1974)* and Bayesian (*Schwarz, 1978*) information criteria, defined

$$AIC := 2k - 2\sum_i \ln L_i \tag{4}$$

$$BIC := \ln(N^{\text{days}})\,k - 2\sum_i \ln L_i, \tag{5}$$

respectively. The number of free parameters $k$ is defined as the number of countries satisfying the criteria for a sequence to be analysable ("SARS-CoV-2 Infection Data"), since there is one free parameter allowed to vary individually for each country. The number of data points for $BIC$ is set to the total number of days $N^{\text{days}}$ in the sequences over all $k$ countries. The $\phi'_i$ model, and the logarithmic median and negative binomial alternatives, each have the same values of $k$ and $N^{\text{days}}$. The 2-day and 3-day alternatives can be expected to have slightly smaller numbers of countries $k$ whose sequences satisfy the analysis criteria, and much smaller numbers $N^{\text{days}}$ of days, since in reality these no longer represent single days. The maximum likelihoood is defined $L_i := P_i^{\text{KS}}$, *i.e.* the Kolmogorov–Smirnov

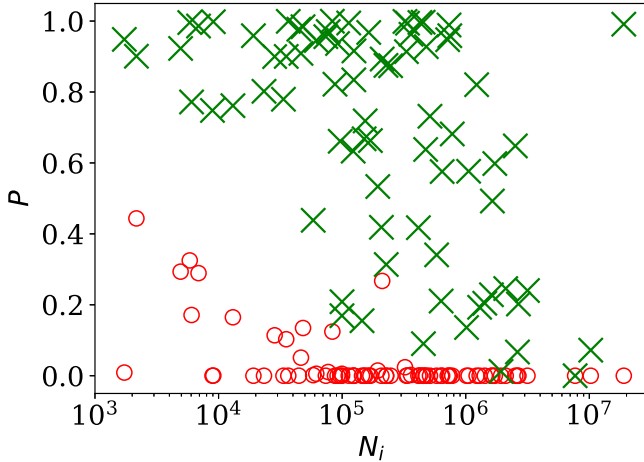

**Figure 2** **Probability of the noise in the country-level daily SARS-CoV-2 counts being consistent with a Poisson point process,** $P_i^{\text{Poiss}}$, **shown as red circles; and probability** $P_i^{\text{KS}}(\phi_i)$ **for the** $\phi_i$ **clustering model proposed here ("Poissonian and** $\phi_i'$ **Models: Full Sequences"), shown as green X symbols;** *versus* $N_i$, **the total number of officially recorded infections for that country.** The horizontal axis is logarithmic. As discussed in the text ("Full Infection Count Sequences"), the Poisson point process is unrealistic for most of these data, while the $\phi_i$ clustering model is consistent with the data for most countries. Plain-text table: zenodo.4765705/phi N full.dat.

probability that the observed values for the country are drawn from a rescaled Poisson (or negative binomial) distribution, as defined above.

# RESULTS

## Data

The 139 countries and territories in the C19CCTF counts data have 27 negative values out of the total of 36,445 values. These can reasonably be interpreted as corrections for earlier overcounts, and we reset these values to zero, with a negligible reduction in the amount of data. Consecutive sequences of days satisfying the criteria listed in "SARS-CoV-2 Infection Data" were found for $M^{\text{valid}} = 78$ countries.

## Clustering of SARS-CoV-2 counts

### *Full infection count sequences*

Figure 2 shows, unsurprisingly, that only a small handful of the countries' daily SARS-CoV-2 counts sequences have noise whose statistical distribution is consistent with the Poisson distribution, in the sense modelled here: $P_i^{\text{Poiss}}$ (red circles) is close to zero in most cases. Specifically, 63 countries (80.8%) are inconsistent with the Poisson distribution at a significance of $P_i^{\text{Poiss}} < 0.01$ and 66 countries (84.6%) are non-Poissonian at $P_i^{\text{Poiss}} < 0.05$. On the contrary, the introduction of the $\phi_i'$ parameter, optimised to $\phi_i$ for each country $i$, provides a sufficiently good fit in most cases, especially for the countries with low clustering $\phi_i$. While some of the probabilities ($P_i^{\text{KS}}(\phi_i)$, green X symbols) in Fig. 2 are low in countries with the highest numbers of infections, these countries also have high $\phi_i$, so are not of interest as indicators of the absence of noise. Among countries with $\phi_i < 10.0$, the lowest probability $P_i^{\text{KS}}$ is that of Algeria with $P_i^{\text{KS}} = 0.17$, *i.e.*, the $\phi_i$ model is

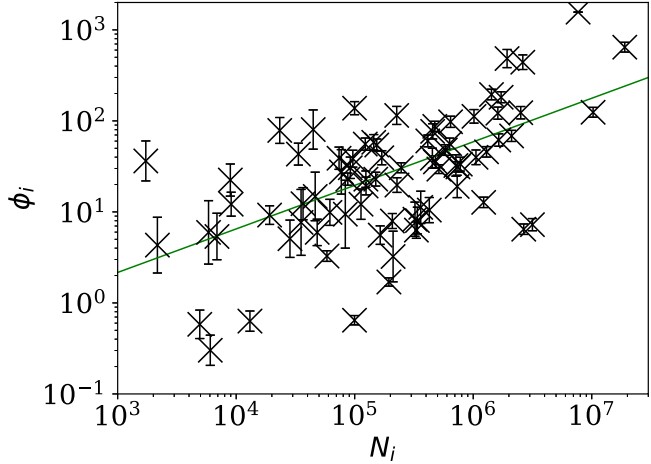

**Figure 3 Noisiness in daily SARS-CoV-2 counts, showing the clustering parameter $\phi_i$ ("Poissonian and $\phi_i'$ Models: Full Sequences") that best models the noise, *versus* the total number of counts for that country $N_i$.** The error bars show standard errors derived from numerical simulations based on the model. The axes are logarithmic, as indicated. Values of the clustering parameter $\phi_i$ below unity indicate sub-Poissonian behaviour—the counts in these cases are less noisy than expected for Poisson statistics. A robust (*Theil, 1950*; *Sen, 1968*) linear fit of $\log^{10} \phi_i$ against $\log^{10} N_i$ is shown as a thick green line ("Full Infection Count Sequences"). Plain-text table: zenodo.4765705/phi N full.dat.

consistent with the data. In contrast, the negative binomial model $\phi_i^{\mathrm{NB}}$ (see "Alternative Analyses" below), which is super-Poissonian by definition, and cannot model sub-Poissonian behaviour, yields $P_i^{\mathrm{KS}} = 0.01$ for Algeria. Consistently with this, the Poissonian model for Algeria gives $P_i^{\mathrm{Poiss}} = 0.005$. The full sequence for Algeria is only fit by the $\phi_i'$ model, which allows sub-Poissonian behaviour.

The consistency of the $\phi_i$ model with most of the data justifies continuing to Fig. 3, which clearly shows a scaling relation: countries with greater overall numbers $N_i$ of infections also tend to have greater noise in the daily counts $n_i(j)$. A Theil–Sen linear fit (*Theil, 1950*; *Sen, 1968*) to the relation between $\log_{10}\phi_i$ and $\log_{10}N_i$ has a zeropoint of $-1.10 \pm 0.44$ and a slope of $0.48 \pm 0.07$, where the standard errors (68% confidence intervals if the distribution is Gaussian) are conservatively generated for both slope and zeropoint by 100 bootstraps. By using a robust estimator, the low $\phi_i$ cases, which appear to be outliers, have little influence on the fit. The fit is shown as a thick green line in Fig. 3.

This $\phi_i$–$N_i$ relation is consistent with $\phi_i \propto \sqrt{N_i}$. To adjust the $\phi_i$ clustering value to take into account the dependence on $N_i$, and given that the slope is consistent with this simple relation, we propose an empirical definition of a normalised clustering parameter

$$\psi_i := \phi_i / \sqrt{N_i}, \qquad (6)$$

so that $\psi_i$ should, by construction, be approximately constant. While the estimated slope of the relation could be used rather than this half-integer power relation, the fixed relation in Eq. (6) offers the benefit of simplicity.

Peer⅃

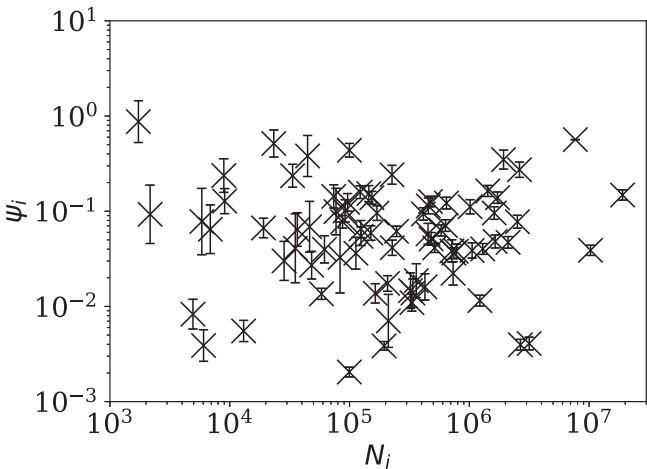

**Figure 4 Normalised noisiness $\psi_i$ (Eq. (6)) for daily SARS-CoV-2 counts *versus* total counts $N_i$.** The error bars are as in Fig. 3, assuming no additional error source contributed by $N_i$. The axes are logarithmic. Several low $\psi_i$ values appear to be outliers of the $\psi_i$ distribution.

This relation should not be confused with the usual Poisson error. By the divisibility of the Poisson distribution, the relation $\phi_i \propto \sqrt{N_i}$ that was found here can be used to show that

$$\sigma[\widehat{\mu}_i(j)/\phi_i] \sim \sqrt{\widehat{\mu}_i(j)/\phi_i}$$

$$\Rightarrow \sigma[\widehat{\mu}_i(j)] \sim \phi_i \sqrt{\widehat{\mu}_i(j)/\phi_i} \propto N_i^{1/4} \widehat{\mu}_i(j)^{1/2}, \tag{7}$$

where $\sigma[x]$ is the standard deviation of random variable $x$. If we accept $\widehat{\mu}_i(j)$ as a fair model for $n_i(j)$ and that $n_i(j)$ is proportional to $N_i$, then we obtain

$$\sigma[n_i(j)] \propto n_i^{3/4}. \tag{8}$$

Figure 4 shows visually that $\psi_i$ appears to be scale-independent, in the sense that the dependence on $N_i$ has been cancelled, by construction. The countries with the 10 lowest values of $\psi_i$ are Algeria, Belarus, Nicaragua, Turkey, Russia, Tajikistan, Croatia, Syria, Saudi Arabia, and Iran. Detailed SARS-CoV-2 daily count noise characteristics for the countries with lowest $\phi_i$ and $\psi_i$ are listed in Table 1, including the Kolmogorov–Smirnov probability that the data are drawn from a Poisson distribution, $P_i^{\text{Poiss}}$, the probability of the optimal $\phi_i$ model, $P_i^{\text{KS}}$, and $\phi_i$ and $\psi_i$.

The approximate proportionality of $\phi_i$ to $\sqrt{N_i}$ for the full sequences is strong and helps separate low-noise SARS-CoV-2 count countries from those following the main trend. However, the results for subsequences shown below in "Subsequences of Infection Counts" suggest that this $N_i$ dependence may be an effect of the typically longer durations of the pandemic in countries where the overall count is higher.

**Table 1  Clustering parameters for the countries with the 10 lowest $\phi_i$ and 10 lowest $\psi_i$ values, *i.e.* the least noise; extended version of table: zenodo.4765705/phi_N_full.dat.**

| Country | $\phi_i'$ Model | | | | | Alternative analyses | | | |
|---|---|---|---|---|---|---|---|---|---|
| | | | | | | $\widehat{v}_i$ | | $\omega_i$ | |
| | $N_i$ | $P_i^{\text{Poiss}}$ | $P_i^{\text{KS}}$ | $\phi_i$ | $\psi_i$ | $P_i^{\text{KS}}$ | $\phi_i$ | $P_i^{\text{KS}}$ | $\omega_i$ |
| Nicaragua | 6,046 | 0.17 | 0.77 | 0.30 | 0.003 | 0.66 | 0.30 | 0.17 | 0.00 |
| Syria | 4,931 | 0.29 | 0.92 | 0.58 | 0.008 | 0.92 | 0.58 | 0.29 | 0.00 |
| Tajikistan | 13,062 | 0.17 | 0.76 | 0.63 | 0.005 | 0.78 | 0.67 | 0.16 | 0.00 |
| Algeria | 99,610 | 0.01 | 0.17 | 0.65 | 0.002 | 0.13 | 0.62 | 0.01 | 0.00 |
| Belarus | 194,284 | 0.01 | 0.53 | 1.70 | 0.003 | 0.40 | 1.57 | 0.46 | 0.58 |
| Croatia | 210,837 | 0.27 | 0.89 | 3.24 | 0.007 | 0.89 | 3.24 | 0.70 | 1.02 |
| Albania | 58,316 | 0.00 | 0.44 | 3.27 | 0.013 | 0.41 | 3.27 | 0.30 | 1.80 |
| New Zealand | 2,164 | 0.44 | 0.90 | 4.32 | 0.092 | 0.94 | 4.32 | 0.86 | 1.19 |
| Australia | 28,430 | 0.11 | 0.90 | 5.07 | 0.030 | 0.90 | 5.69 | 0.87 | 3.55 |
| Thailand | 6,884 | 0.29 | 0.99 | 5.37 | 0.064 | 0.99 | 5.37 | 0.96 | 3.80 |
| Algeria | 99,610 | 0.01 | 0.17 | 0.65 | 0.002 | 0.13 | 0.62 | 0.01 | 0.00 |
| Belarus | 194,284 | 0.01 | 0.53 | 1.70 | 0.003 | 0.40 | 1.57 | 0.46 | 0.58 |
| Nicaragua | 6,046 | 0.17 | 0.77 | 0.30 | 0.003 | 0.66 | 0.30 | 0.17 | 0.00 |
| Turkey | 2,669,568 | 0.00 | 0.20 | 6.46 | 0.003 | 0.16 | 6.09 | 0.16 | 5.07 |
| Russia | 3,159,297 | 0.00 | 0.24 | 7.24 | 0.004 | 0.19 | 7.08 | 0.22 | 6.03 |
| Tajikistan | 13,062 | 0.17 | 0.76 | 0.63 | 0.005 | 0.78 | 0.67 | 0.16 | 0.00 |
| Croatia | 210,837 | 0.27 | 0.89 | 3.24 | 0.007 | 0.89 | 3.24 | 0.70 | 1.02 |
| Syria | 4,931 | 0.29 | 0.92 | 0.58 | 0.008 | 0.92 | 0.58 | 0.29 | 0.00 |
| Saudi Arabia | 331,359 | 0.00 | 0.91 | 6.31 | 0.010 | 0.84 | 6.17 | 0.83 | 4.90 |
| Iran | 1,225,142 | 0.00 | 0.82 | 12.73 | 0.011 | 0.58 | 11.61 | 0.71 | 11.35 |

### Subsequences of infection counts

Figures 5–7 show the equivalent of Fig. 2 for sequences of lengths 28, 14 and 7 days, respectively. The Theil–Sen robust fits to the logarithmic $(\phi_i^{28}, N_i)$; $(\phi_i^{14}, N_i)$; and $(\phi_i^{7}, N_i)$ relations are zeropoints and slopes of 0.57 ± 0.43 and 0.06 ± 0.08; 0.52 ± 0.47 and 0.01 ± 0.09 ; and −0.10 ± 0.83 and 0.02 ± 0.13, respectively. There is clearly no significant dependence of $\phi_i^d$ on $N_i$ for any of these fixed length subsequences, in contrast to the case of the $\phi_i$ dependence on $N_i$ for the full count sequences. Thus, the empirical motivation for using $\psi_i$ (Eq. (6)) to discriminate between the countries' full sequences of SARS-CoV-2 data is not justified from the information gained from the subsequences alone. Tables 2–4 show the countries with the least noisy sequences as determined by $\phi_i^{28}$, $\phi_i^{14}$ and $\phi_i^{7}$, respectively.

Tables 2 and 3 show that the lists of countries with the strongest anti-clustering are similar to one another. Thus, Fig. 8 shows the SARS-CoV-2 counts curves for countries with the lowest $\phi_i^{28}$, and Fig. 9 the curves for those with the lowest $\phi_i^{7}$. Both figures exclude countries with total counts $N_i \leq 10000$, in which low total counts tend to give low clustering. It is clear in these figures that several countries have subsequences that are strongly sub-Poissonian—with some form of anti-clustering, whether natural or artificial.

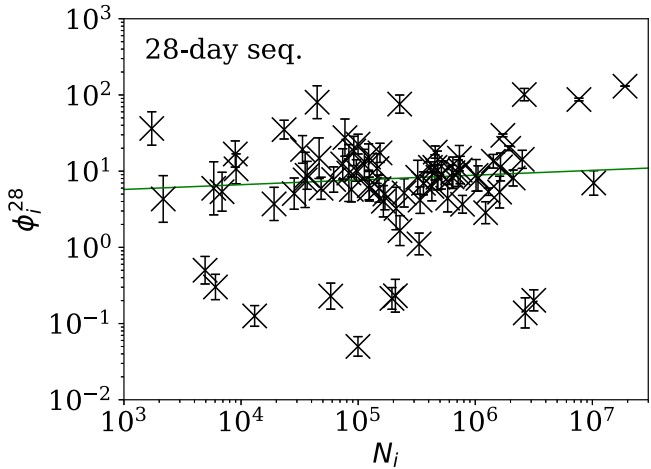

**Figure 5** Clustering parameter $\phi_i^{28}$ for the 28-day sequence of lowest $\phi_i^{28}$, as in Fig. 3. The vertical axis range is expanded from that in Fig. 3, to accommodate lower values. A robust (*Theil, 1950*; *Sen, 1968*) linear fit of $\phi_i^{28}$ against $\log_{10}N_i$ is shown as a thick green line ("Full Infection Count Sequences"). Plain-text table: zenodo.4765705/phi_N_28days.dat.

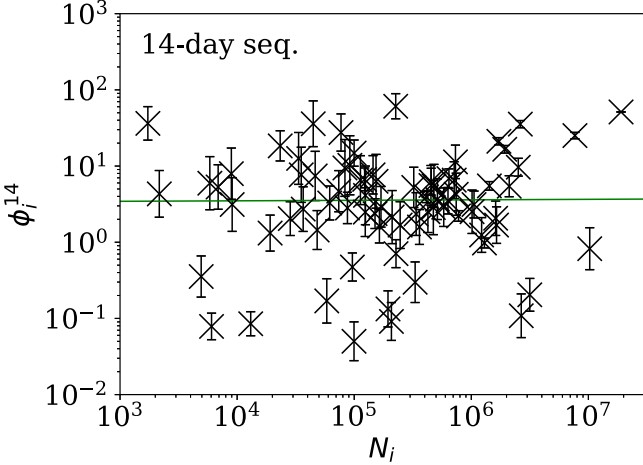

**Figure 6** Clustering parameter $\phi_i^{14}$ for the 14-day sequence of lowest $\phi_i^{14}$, as in Fig. 5. plain-text table: zenodo.4765705/phi_N_14days.dat.

Countries in the median of the $\phi_i^{28}$ and $\phi_i^{7}$ distributions have their curves shown in Fig. 6 for comparison. It is visually clear in the figure that the counts are dispersed widely beyond the Poissonian band, and that the $\phi_i^{28}$ and $\phi_i^{7}$ models are reasonable as a model for representing about 68% of the counts within one standard deviation of the model values.

### Alternative analyses

Figure 11 (left) shows that the logarithmic median model ("Logarithmic Median Model") of the counts gives almost identical best estimates to those of the primary model, *i.e.* $\psi_i^{\text{LM}} \approx \psi_i$, but Table 5 shows very strong evidence favouring the original, arithmetic median model.
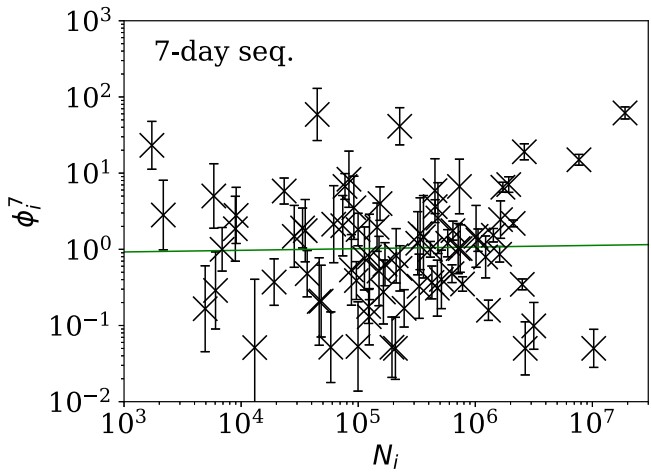

**Figure 7 Clustering parameter $\phi_i^7$ for the 7-day sequence of lowest $\phi_i^7$, as in Fig. 5.** There are clearly a wider overall scatter and bigger error bars compared to Figs. 5 and 6; a low $\phi_i^7$ is a noisier indicator than $\phi_i^{28}$ and $\phi_i^{14}$ for individual countries. Plain-text table: zenodo.4765705/phi_N_07days.dat.

**Table 2 Least noisy 28-day sequences—clustering parameters for the countries with the 10 lowest $\phi_i^{28}$ values; extended table: zenodo.4765705/phi_N_28days.dat.**

| Country | $N_i$ | $\langle n_i^{28} \rangle$ | $P_i^{\text{Poiss}}$ | $P_i^{\text{KS}}$ | $\phi_i^{28}$ | Starting date |
|---|---|---|---|---|---|---|
| Algeria | 99,610 | 227.6 | 0.00 | 0.36 | 0.05 | 2020-09-03 |
| Tajikistan | 13,062 | 63.0 | 0.02 | 0.96 | 0.13 | 2020-06-07 |
| Turkey | 2,669,568 | 1,014.5 | 0.03 | 1.00 | 0.14 | 2020-06-30 |
| Russia | 3,159,297 | 5,403.8 | 0.26 | 0.59 | 0.20 | 2020-07-20 |
| Belarus | 194,284 | 921.9 | 0.14 | 0.89 | 0.21 | 2020-05-08 |
| Albania | 58,316 | 203.8 | 0.33 | 0.64 | 0.23 | 2020-09-27 |
| United Arab Emirates | 207,822 | 512.8 | 0.08 | 0.23 | 0.23 | 2020-04-14 |
| Nicaragua | 6,046 | 135.7 | 0.17 | 0.77 | 0.30 | 2020-07-07 |
| Syria | 4,931 | 70.0 | 0.19 | 0.91 | 0.50 | 2020-08-15 |
| Saudi Arabia | 331,359 | 1,182.2 | 0.47 | 0.54 | 1.11 | 2020-04-12 |

Figure 11 (right) shows that the negative binomial model ("Negative Binomial Model") roughly gives $\psi_i^{\text{NB}} \sim \psi_i$ (*i.e.* $\omega_i \sim \phi_i$), tending to $\psi_i^{\text{NB}} < \psi_i$, especially for the least clustered cases. The error bars are very big for $\psi_i^{\text{NB}}$ for several countries. Table 5 again shows very strong evidence favouring the original model over the negative binomial model.

Figure 12 shows that the counts grouped (summed) in pairs and triplets ("Does Anti-Clustering Exist in Grouped Data?") yield $\psi_i^{2d}$ and $\psi_i^{3d}$ with more scatter and generally larger error bars than that of $\psi_i$, and $\psi_i^{2d}$ and $\psi_i^{3d}$ are mostly greater than $\psi_i$. Whether the AIC and BIC evidence (Table 5) for 2- and 3-day grouped data can be directly compared to that of the main analysis depends on whether the grouped data can be considered to be the same observational data as the original data, modelled with fewer free parameters. Since the characteristic of study is the noise, not the signal, the validity of this

**Table 3 Least noisy 14-day sequences—clustering parameters for the countries with the 10 lowest $\phi_i^{14}$ values; extended version of table: zenodo.4765705/phi_N_14days.dat.**

| Country | $N_i$ | $\langle n_i^{14} \rangle$ | $P_i^{Poiss}$ | $P_i^{KS}$ | $\phi_i^{14}$ | Starting date |
|---------|-------|------------|----------|--------|----------|---------------|
| Algeria | 99,610 | 285.9 | 0.12 | 0.40 | 0.05 | 2020-09-01 |
| Nicaragua | 6,046 | 73.6 | 0.12 | 0.98 | 0.08 | 2020-09-22 |
| Tajikistan | 13,062 | 64.6 | 0.02 | 0.99 | 0.09 | 2020-06-11 |
| United Arab Emirates | 207,822 | 521.2 | 0.11 | 0.56 | 0.09 | 2020-04-19 |
| Turkey | 2,669,568 | 971.6 | 0.12 | 0.86 | 0.11 | 2020-07-08 |
| Belarus | 194,284 | 945.6 | 0.22 | 1.00 | 0.13 | 2020-05-12 |
| Albania | 58,316 | 143.4 | 0.21 | 0.96 | 0.17 | 2020-09-01 |
| Russia | 3,159,297 | 5627.0 | 0.47 | 0.98 | 0.20 | 2020-07-21 |
| Saudi Arabia | 331,359 | 1227.5 | 0.38 | 0.96 | 0.30 | 2020-04-19 |
| Syria | 4,931 | 76.6 | 0.42 | 0.96 | 0.35 | 2020-08-14 |

**Table 4 Least noisy 7-day sequences—clustering parameters for the countries with the 10 lowest $\phi_i^7$ values; extended table: zenodo.4765705/phi_N_07days.dat.**

| Country | $N_i$ | $\langle n_i^7 \rangle$ | $P_i^{Poiss}$ | $P_i^{KS}$ | $\phi_i^7$ | Starting date |
|---------|-------|----------|----------|--------|---------|---------------|
| United Arab Emirates | 207,822 | 544.9 | 0.24 | 0.99 | 0.05 | 2020-04-27 |
| India | 10,266,674 | 10109.3 | 0.34 | 0.60 | 0.05 | 2020-06-06 |
| Turkey | 2,669,568 | 929.6 | 0.22 | 0.93 | 0.05 | 2020-07-15 |
| Tajikistan | 13,062 | 51.9 | 0.16 | 0.77 | 0.05 | 2020-06-28 |
| Albania | 58,316 | 297.7 | 0.23 | 0.98 | 0.05 | 2020-10-18 |
| Belarus | 194,284 | 947.9 | 0.60 | 0.94 | 0.05 | 2020-05-13 |
| Algeria | 99,610 | 204.3 | 0.37 | 0.49 | 0.05 | 2020-10-14 |
| Russia | 3,159,297 | 5076.7 | 0.36 | 0.68 | 0.10 | 2020-08-09 |
| Ethiopia | 124,264 | 456.7 | 0.83 | 0.93 | 0.13 | 2020-12-13 |
| Poland | 1,294,878 | 297.7 | 0.31 | 0.96 | 0.16 | 2020-06-20 |

direct comparison is doubtful. Nevertheless, if the values of the AIC and BIC evidence are considered literally, then the 2-day grouping would yield a worse model than the model of the daily data, while the 3-day grouping would yield a better model than that for the daily data. The comparison of these different analyses could potentially be used to obtain a deeper understanding of the complex dynamics of this pandemic. The epidemiologically relevant sociological parameters of countries around the world are highly diverse (varying in population density, patterns of social contact, tendency to obey or disobey official health guidelines such as lockdown measures, demographic profiles, quality and availability of health services, communication patterns, frequency of COVID-19 comorbidity conditions, climate (*Afshordi et al., 2020*)), so comparison of the clustering behaviour on these different time scales might help to separate out some of these contributions.

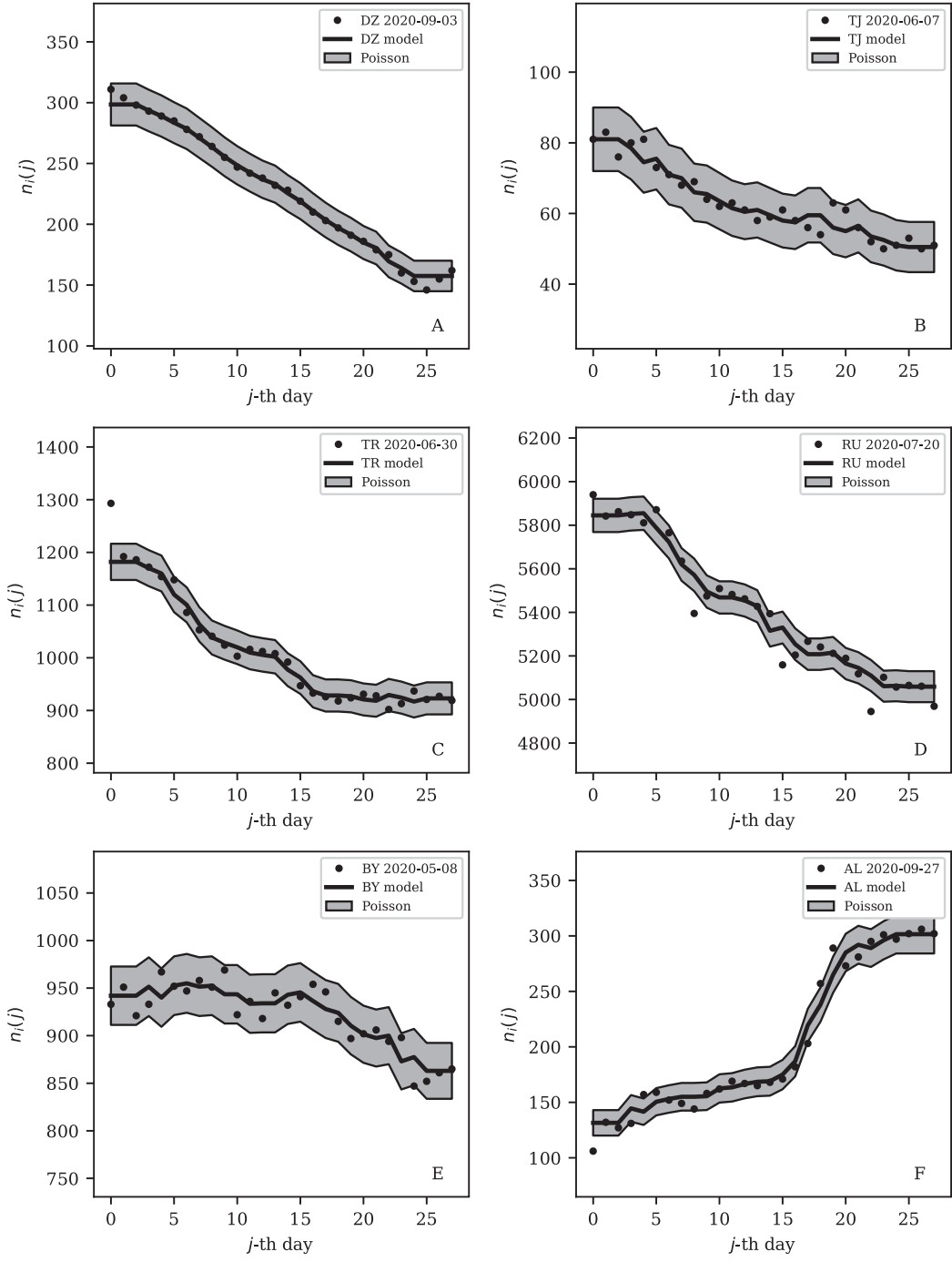

**Figure 8 Least noisy 28-day official SARS-CoV-2 national daily counts for countries with total counts $N_i > 10,000$ (see Fig. 5 and Table 2), shown as dots in comparison to the $\widehat{\mu}_i(j)$ model (median of the four neighbouring days) and 68% error band for the Poisson point process.** The ranges in daily counts (vertical axis) are chosen automatically and in most cases do not start at zero. About nine (32%) of the points should be outside of the shaded band unless the counts have an anti-clustering effect that weakens Poisson noise. The dates indicate the start date of each sequence. ISO-3166-1 key: (A) DZ: Algeria; (B) TJ: Tajikistan; (C) TR: Turkey; (D) RU: Russia; (E) BY: Belarus; (F) AL: Albania.

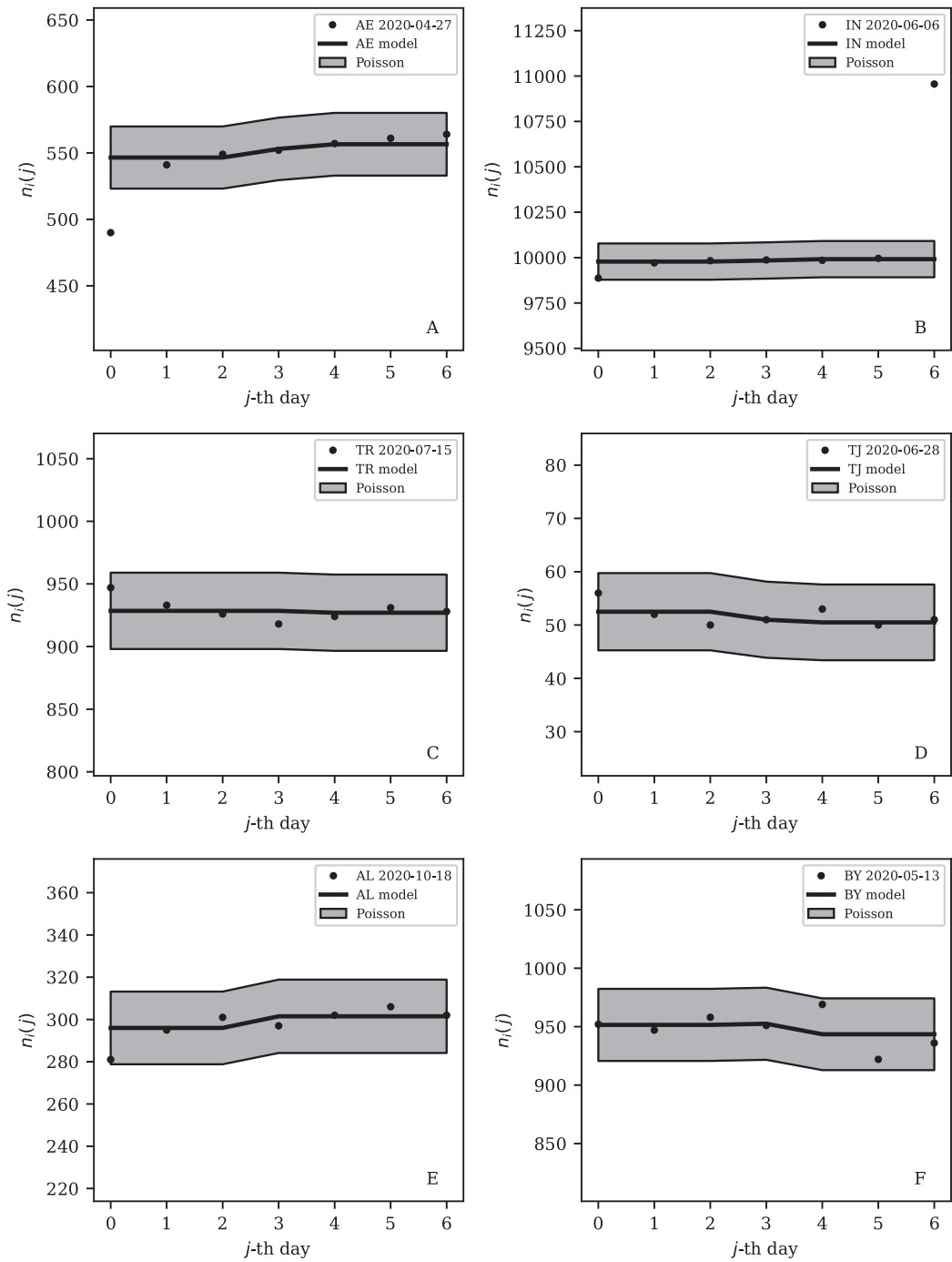

**Figure 9 Least noisy 7-day daily counts for countries with total counts $N_i > 10,000$ (see Fig. 7 and Table 4), as in Fig. 8.** A concentration of points close to the model indicates an anti-clustering effect; about 68% (five) of the points should scatter up and down throughout the shaded band if the counts are Poissonian, and about 32% (two) should be outside the band. In several cases, the data points appear to be mostly stuck to the model, with almost no scatter. ISO-3166-1 key: (A) AE: United Arab Emirates; (B) IN: India; (C) TR: Turkey; (D) TJ: Tajikistan; (E) AL: Albania; (F) BY: Belarus.

## Comparison with the RSF press freedom index

Figures 13–16 show the relation between $\phi_i$ and $\psi_i$ and the RSF Press Freedom Index (PFI[202]; "RSF Press Freedom Index") for the full sequences and subsequences. Table 6 non-parametrically tests for correlations in these relations using the Kendall rank correlation statistic $\tau$ (*Kendall, 1938*; *Kendall, 1970*; *Croux & Dehon, 2010*). The first row of the table shows that the unnormalised clustering parameter $\phi_i$ for the full sequence and subsequences generally anticorrelates with PFI[2020]. The strongest case is for 7-day subsequences, in which case the anticorrelation is significant at $P^\tau = 0.0108$.

The normalised clustering parameter $\psi_i$ was found to be necessary above (Eq. (6)) to remove dependence on the total infection scale $N_i$ in the full sequences. The second row of Table 6 shows that for $\psi_i$, the anticorrelation is significant at the $P^\tau < 0.05$ level for the full sequence ($P^\tau = 0.0408$) and for all the subsequences. However, the analysis of the subsequence results ("Subsequences of Infection Counts") only justifies considering $\psi_i$ as the preferred parameter for the full sequence, and using $\phi_i^{28}, \phi_i^{14}$, and $\phi_i^7$ for the subsequences. Together, $\psi_i, \phi_i^{28}, \phi_i^{14}$, and $\phi_i^7$ yield a median significance level of $P^\tau = 0.0496 < 0.05$ (the significance is stronger in the JHU CSSE data; see the corresponding table in Appendix A). Thus, there is statistically significant evidence that the worse the press freedom is in a country (as measured by higher PFI[2020]), the more likely it is that the SARS-CoV-2 daily counts are best modelled as sub-Poissonian.

This result is an anticorrelation; it is not proof of a causal relation. Nevertheless, a simple explanation of the observed relation would be that there is interference in the data in association with a lack of media freedom.

## DISCUSSION

Figures 3–7 vary in the degree to which they separate some groups of countries as being unusual in terms of the characteristics of their location in the $(N_i, \psi_i)$ plane. On visual inspection, Fig. 5, for $\phi_i^{28}$, appears to show the sharpest division between the main relation between clustering and total infection count, in which nine countries appear to have sub-Poissonian preferred models in a group well-separated from the others. If we interpret the sub-Poissonian behaviour as a result of interference associated with the lack of media freedom (high PFI[2020], "Comparison with the RSF Press Freedom Index", Table 6), then the more significant results are those for $\phi_i^7$ (Fig. 7, Table 4). If interference did occur, then other public evidence of interference might add credibility to the interpretation. Here, some possible interpretations are discussed, including some individual low-noise sequences in Figs. 8 and 9. Some typical sequences (as selected by median $\phi_i^{28}$ and $\phi_i^7$) are shown for comparison in Fig. 10.

The analysis in this paper makes very few assumptions and does not claim to measure the full nature of the pandemic. The following interpretations of the numerical results would benefit from future studies that attempt worldwide models of the underlying epidemiology of the pandemic. Detailed modelling is usually restricted to a small number of countries (*e.g. Chowdhury et al., 2020*; *Kim et al., 2020*; *Molina-Cuevas, 2020*; *Jiang, Zhao & Shao, 2021*; *Afshordi et al., 2020*).
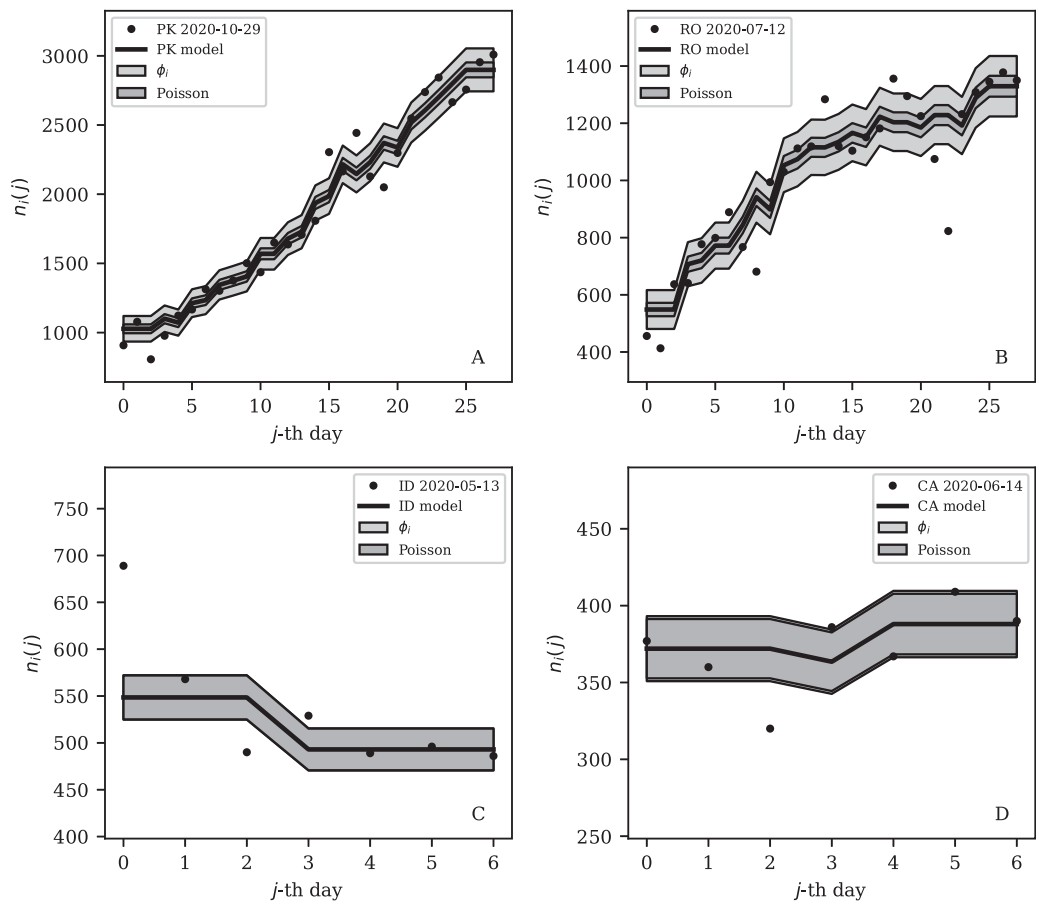

**Figure 10** Typical (median) 28-day (above) and 7-day (below) daily counts, as in **Figs. 8** and **9**. The dark shaded band again shows a Poissonian noise model, which underestimates the noise. A faint shaded band shows the $\phi_i$ models for these countries' SARS-CoV-2 daily counts, and should contain about 68% of the infection count points. ISO-3166-1 key: (A) PK: Pakistan; (B) RO: Romania; (C) ID: Indonesia; (D) CA: Canada.               

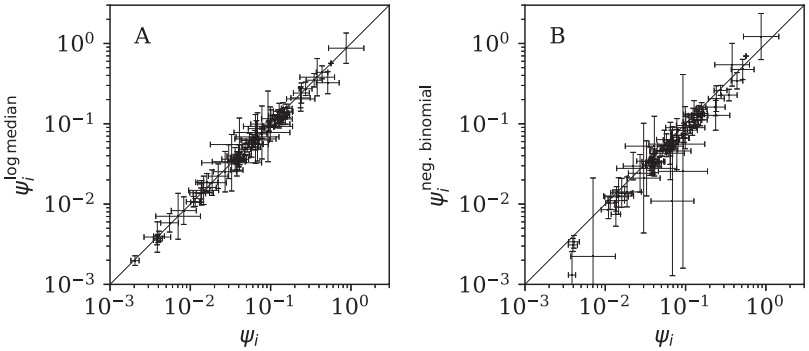

**Figure 11** (A) Normalised clustering parameter $\psi_i^{\mathrm{LM}}$ (**Eq. (6)**) using the logarithmic median model of the expected full-sequence counts ("Logarithmic Median Model") *versus* $\psi_i$ for the primary analysis. (B) Normalised clustering $\psi_i^{NB} := \omega_i/\sqrt{N_i}$ for the negative binomial model (see **Eqs. (2), (3)**) versus $\psi_i$. A line $\psi_i^{LM} = \psi_i$ shows and $\psi_i^{NB} = \psi_i$, respectively. The data point for Algeria, with $\log_{10}\psi_i = -2.69 \pm 0.05$, $\log_{10}\psi_i^{NB} = -5.69 \pm 0.93$, lies below the displayed area in the right-hand panel. Plain-text table: zenodo. 4765705/phi_N_full.dat.               

**Table 5** *Akaike (1974)* and Bayesian (*Schwarz, 1978*) information criteria for the $\phi_i'$ and alternative analyses; plain-text version: **zenodo.4765705/AIC_BIC_full.dat**.

| Model | $\phi_i'$ | | Log. median | | Neg. binomial | | 2-Day grouping | | 3-Day grouping | |
|---|---|---|---|---|---|---|---|---|---|---|
| | *AIC* | *BIC* | *AIC* | *BIC* | *AIC* | *BIC* | *AIC* | *BIC* | *AIC* | *BIC* |
| | 268.60 | 848.87 | 289.91 | 870.18 | 377.52 | 957.79 | 313.21 | 878.50 | 208.49 | 743.75 |

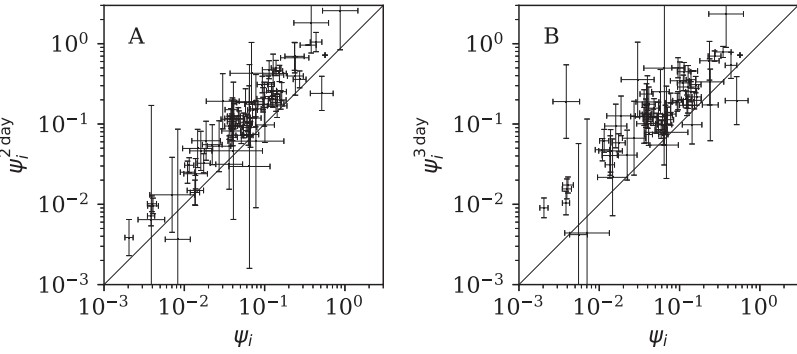

**Figure 12 Normalised noisiness $\psi_i^{2d}$ and $\psi_i^{3d}$ (Eq. (6)) for counts summed in successive pairs (A) and triplets (B) of days, respectively, versus that for the primary analysis.** A line shows $\psi_i^{2d} = \psi_i$ and $\psi_i^{3d} = \psi_i$, respectively. Plain-text table: zenodo.4765705/phi_N_full.dat.

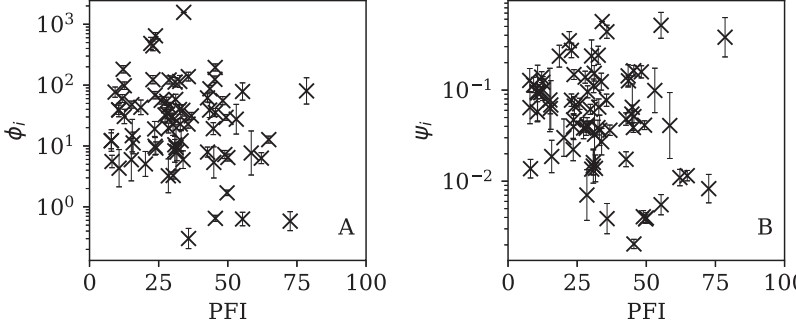

**Figure 13 Dependence of $\phi_i$ (*left*: A) and $\psi_i$ (*right*: B) on the Press Freedom Index (PFI$^{2020}$) for the full sequences.** The vertical axis ranges in these two panels and through to Fig. 16 differ from one another.

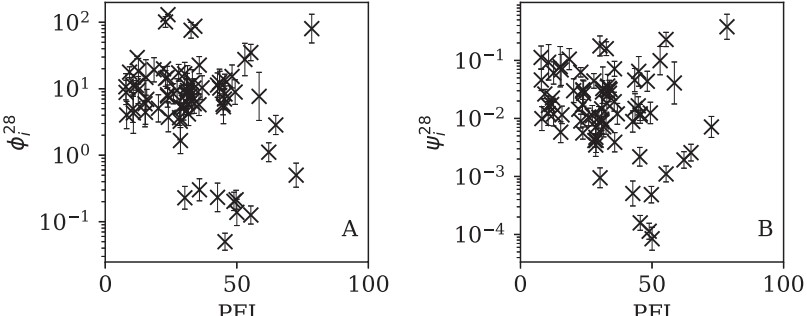

**Figure 14 Dependence of $\phi_i^{28}$ (A) and $\psi_i^{28}$ (B) on PFI$^{2020}$ for the 28-day subsequences.**

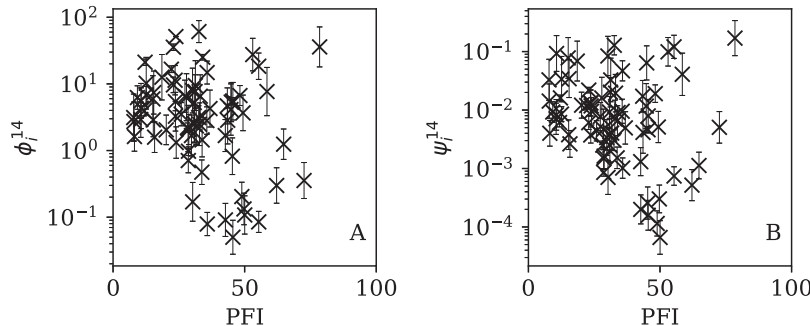

**Figure 15 Dependence of $\phi_i^{14}$ (A) and $\psi_i^{14}$ (B) on PFI$^{2020}$ for the 14-day subsequences.**

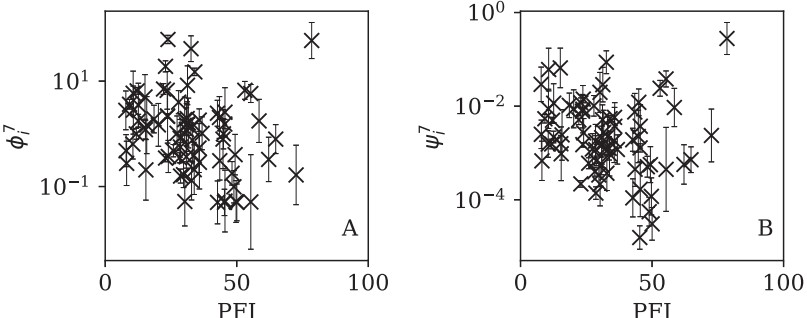

**Figure 16 Dependence of $\phi_i^7$ (A) and $\psi_i^7$ (B) on PFI$^{2020}$ for the 7-day subsequences.**

**Table 6 Kendall $\tau$ statistic and its significance (two-sided) $P^\tau$ for the null hypothesis of no correlation between the ranking of PFI$^{2020}$ and $\phi_i$ or $\psi_i$ for the full data or subsequences; plain-text version: zenodo.4765705/pfi_correlations_table.dat.**

| Parameter | Full | | 28-day | | 14-day | | 7-day | |
|---|---|---|---|---|---|---|---|---|
| | $\tau$ | $P^\tau$ | $\tau$ | $P^\tau$ | $\tau$ | $P^\tau$ | $\tau$ | $P^\tau$ |
| $\phi_i$ | −0.118 | 0.131 | −0.126 | 0.108 | −0.148 | 0.0584 | −0.200 | 0.0108 |
| $\psi_i$ | −0.160 | 0.0408 | −0.157 | 0.0445 | −0.176 | 0.0249 | −0.170 | 0.0300 |

## High total infection count

While the main question of interest in this paper is whether anti-clustering can be detected, the results may also indicate characteristics of countries with high clustering values. The United States, India and Brazil are clearly separated in Figs. 3 and 4 from the majority of other countries by their high official total infection counts of about $10^7$. They have correspondingly higher clustering values $\phi_i$, although their normalised clustering values $\psi_i$ are in the range of about $0.01 < \psi_i < 1$ covered by the majority of countries in Fig. 4.

It does not seem realistic that the $\phi_i$ values greater than 600 for the US and Brazil are purely an effect of intrinsic infection events—superspreader events in crowded places or nursing homes. While individual big clusters may occur given the high overall scale of infections, it seems more likely that there is a strong role played by administrative clustering. Both countries are federations, and have numerous geographic administrative

subdivisions with a diversity of political and administrative methods. A plausible explanation for the dominant effect yielding $\phi_i > 600$ in these two countries is that on any individual day, the arrival and full processing of reports depends on a number of sub-national administrative regions, each reporting a few hundred new infections.

For example, if there are 100 reporting regions, with typically about 10 of these each reporting about 600 infections daily, then typically (on about 68% of days) there will be about 7 to 13 reports per day. This would give a range varying from about 4,200 to 7,800 cases per day, rather than 5,923 to 6,077, which would be the case for unclustered, Poissonian counts ( since $\sqrt{6000} \approx 77$). Lacking a system that obliges sub-national divisions—and laboratories—to report their test results in time-continuous fashion and that validates and collates those reports on a time scale much shorter than 24 h, this type of clustering seems natural in the sociological sense. It is also possible that in these two large federations, the intrinsic heterogeneity compared to many countries of smaller populations leads to other noise effects that combine with the administrative effect of stochasticity in the number of regional reports received as sketched above.

India's overall position in the $(\psi_i, N_i)$ plane (Fig. 4 and Table 1) appears quite typical, with an unnormalised clustering parameter $\phi_i = 124.45 \times 10^{\pm0.054}$. However, Table 4 shows that despite its large overall infection count, India achieved a 7-day sequence with a preferred $\phi_i^7 = 0.05$, giving it a place in Table 4 and being easy to identify in the bottom-right part of Fig. 7. Figure 9 presents this subsequence. Five values appear almost exactly on the model curve rather than scattering above and below. Moreover, the value is just below 10,000. Epidemiologically, it is not credible to believe that 10,000 officially reported cases per day should be an attractor resulting from the pattern of infections and system of reporting. Given that the value of 10,000 is a round number in the decimal-based system, a reasonable speculation would be that the daily counts for India were artificially held at just below 10,000 for several days. The crossing of the 10,000 psychological threshold of daily infections was noted in the media (*Porecha, 2020*), but the lack of noise in the counts during the week preceding the crossing of the threshold appears to have gone unnoticed. After crossing the 10,000 threshold, the daily infections in India continued increasing, as can be seen in the full counts (zenodo.org/4765705/WP_C19CCTF_SARSCoV2.dat).

## Neither poissonian nor super-poissonian

The negative binomial model $\phi_i^{\text{NB}}$ ("Alternative Analyses") rejects the possibility of Algeria having a super-Poissonian noise distribution at $P_i^{\text{KS}} = 0.01$. The Poissonian model for Algeria is similarly rejected with $P_i^{\text{Poiss}} = 0.005$. However, the $\phi_i$ model does model the Algeria data adequately, with a modest probability of $P_i^{\text{KS}} = 0.17$.

Figure 8 dramatically shows the least noisy 28-day sequence for Algeria. Only two days of SARS-CoV-2 recorded infections during this period appear to have diverged towards the edge of the Poissonian 68% band, rather than about nine, the expected number that should be outside this band for a Poissonian distribution. Almost all of the points appear to stick extremely closely to the median model. It is difficult to imagine a natural process for obtaining noise that is this strongly sub-Poissonian, especially in the context

where most countries have super-Poissonian daily counts. Compartmental epidemic modelling of the Algerian data, which has been published for the period ending 24 May 2020 (*Rouabah, Tounsi & Belaloui, 2021*), could be used to try to reconstruct the true daily counts.

## Low normalised clustering $\psi_i$ or subsequence clustering $\phi_i^{28}$, $\phi_i^{14}$ or $\phi_i^7$

### Low clustering, high $N_i$

Turkey and Russia have total infection counts of about 3 million, similar to those of several other countries, but have managed to keep their daily infection rates much less noisy—by about a factor of 10 to 100—than would be expected from the general pattern displayed in the figures. These two countries appear as an isolated pair in the bottom-right of both Figs. 4 and 5, and appear in all four tables of low $\psi_i$ (Table 1) and low subsequence $\phi_i$ (Tables 2–4). Russia has the very modest value of $\phi_i = 7.24 \times 10^{\pm 0.067}$ and Turkey has $\phi_i = 6.46 \times 10^{\pm 0.057}$, despite their large total infection counts. This would require that both intrinsic clustering of infection events and administrative procedures work much more smoothly in Russia and Turkey than in other countries with comparable total infection counts. Tables 2 and 3 and Fig. 8 show that the Russian and Turkish official SARS-CoV-2 counts indeed show very little noise compared to more typical cases (Fig. 10). There appear to be weekend dips in the Russian case (see "Weekend Dips in the Counts" below). Since these are included in the analysis, an exclusion of the weekend dips would lead to an even lower clustering estimate. At the intrinsic epidemiological level, if the Russian and Turkish counts are to be considered accurate, then very few clusters—in nursing homes, religious gatherings, bars, restaurants, schools, shops—can have occurred. Moreover, laboratory testing and transmission of data through the administrative chain from local levels to the national health agency must have occurred without the clustering effects that are present in the data for the United States, Brazil, India, and other countries with high total infection counts $N_i > 2$ million, for which $\phi_i$ is typically above 100. International media interest in Russian COVID-19 data has mostly focussed on controversy related to COVID-19 death counts (*Cole, 2020*), with apparently no attention given so far to the modestly super-Poissonian nature of the daily counts, in contrast to the strongly super-Poissonian counts of other countries with high total infection counts. How did Russia and Turkey achieve low $\phi_i$ (super-Poissonian), *i.e.* low clustering?

### Low clustering, medium $N_i$

Some cases of interest appear among the countries with officially lower total infection counts. The Belarus (BY) case is present in all four tables (Tables 1–4). The least noisy Belarusian counts curve appears in Figs. 8 and 9. As with the other panels in the daily counts figures, the vertical axis is set by the data instead of starting at zero, in order to best display the information on the noise in the counts. With the vertical axis starting at zero, the Belarus daily counts would look nearly flat in this figure. They appear to be bounded above by the round number of 1,000 SARS-CoV-2 infections per day, which, again, as in the case of India, could appear to be a psychologically preferred barrier. Media have expressed scepticism of Belarusian COVID-19 related data (*Kramer, 2020*; *AFN, 2020*). The Albanian case (Figs. 8 and 9) also could be interpreted as hitting a psychological

barrier of a decimal round number, an artificial cap of 300 infections per day, in mid-October 2020.

One remaining case of a coincidence is that the lowest noise 7-day sequence listed for Poland (Table 4) is for the 7-day period starting 20 June 2020, with $\phi_i^7 = 0.16 \times 10^{\pm 0.13}$. This is a factor of about 300 below Poland's clustering value for the full sequence of its SARS-CoV-2 daily infection counts, $\phi_i = 45.71 \times 10^{\pm 0.057}$, which Fig. 3 shows is typical for a country with an intermediate total infection count. On 28 June 2020, there was a *de facto* (of disputed constitutional validity, *Wyrzykowski, 2020*; *Letowska & Pacewicz, 2020*) first-round presidential election in Poland. Figure 9 shows that the counts for Poland during the final pre-first-round-election week did not scatter widely throughout the Poissonian band. A decimal-system round number also appears in this figure: the daily infection rate is slightly above about 300 infections per day and drops to slightly below that. This appears to be the same psychological daily infection count attractor as for Albania. The intrinsic clustering of SARS-CoV-2 infections in Poland together with testing and administrative clustering of the confirmed cases appear to have temporarily disappeared just prior to the election date, yielding what is best modelled as an incident of sub-Poissonian counts.

### JHU CSSE data
The JHU CSSE data give mostly similar results to the C19CCTF data. These are presented and briefly discussed in Appendix A.

### Weekend dips in the counts
One sociological contribution to noise not mentioned above is that in several countries, the official daily counts are lower on or immediately after weekends. Credible factors include fewer medical and laboratory workers available to carry out tests and fewer administrators registering, collecting and transmitting data. A dip in the counts on weekends would tend to add noise to the daily count time series, making the above results conservative. These dips can be quantified using the one-dimensional discrete fast Fourier transform (FFT). With the usual FFT convention, we transform $n_i(j)$ into $f_i(j)$ at $j$ days, where $f_i(0)$ is the mean and a weekly dip should appear as a negative value at $f_i(7)$. We define a weekend dip $w_i$ for country $i$ by subtracting the mean of the neighbours and normalising:

$$w_i := 1 + \pi \frac{f_i(7) - [f_i(6) + f_i(8)]/2}{f_i(0)} . \tag{9}$$

This should correspond to a multiplicative factor, *i.e.*, $w_i = 0.85$ means a 15% dip.

Figure 17 shows the distribution of $w_i$ (mean ± std. error: 1.001 ± 0.015; std. dev.: 0.137; median: 0.999; interquartile range: 0.104). Unexpectedly, not only are there several countries with dips, but there are also several countries with a strong *excess* signal on the 7-day time scale. There is no reason to expect the overall distribution to be Gaussian. The Shapiro–Wilk statistic (*Shapiro & Wilk, 1965*) is $W = 0.806$, rejecting the possibility of the distribution being Gaussian to extremely high significance: $p = 9.82 \ 10^{-9}$. Future work

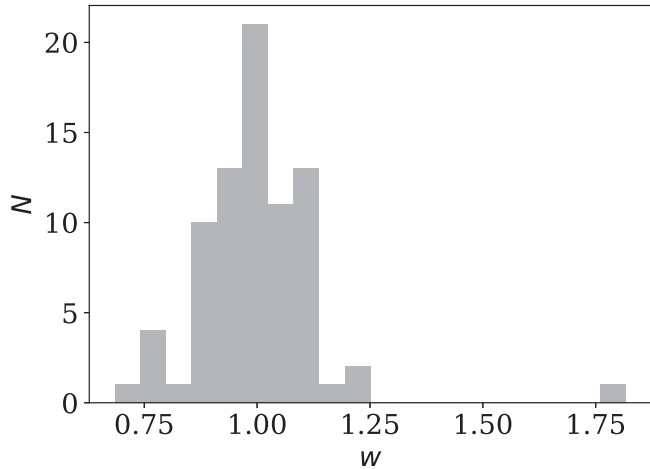

**Figure 17 Histogram of weekly dip $w_i$ (Eq. (9)) in national daily SARS-CoV-2 counts.** Values below unity indicate a dip; values above unity indicate a bump. Plain-text full list of $w_i$: zenodo.4765705/phi_N_full.dat.               

in studying the noise characteristics of a pandemic could take into account this weekly component of daily infection statistics.

## Further statistical models: autoregression

A possible extension of the current work would be to iteratively consider an autoregressive model (*e.g.*, *Papoulis & Pillai, 2002*, §12-3; *Fokianos & Tjøstheim, 2011*; *Agosto et al., 2021*) for each time series. An initial model such as the one used here, the median of the preceding and succeeding days, could first be inferred from the sequence. This would be subtracted from the time series $n_i(j)$ to obtain a process that could be assumed as having a stationary central value and a time-varying noise distribution. An autoregressive model of the resulting sequence (or its logarithm) could then be modelled by a time-dependent ($j$-dependent) Poissonian or negative binomial stochastic term to find the optimal autoregression coefficients. The resulting coefficients could then be used to subtract an improved model from the times series and obtain a new iteration of an autoregression model. Continuing the iteration might lead to convergence on a specific autoregressive model that is stable against further iteration. In this case, the residual noise could then be analysed as in the current work. Alternatively, time series analysis of SARS-CoV-2 counts allowing time-varying trends (*Harvey & Kattuman, 2020*) could be carried out prior to analysing the properties of the noise itself, as in this work.

## RSF press freedom index

Although the relations in Figs. 13–16 generally show anticorrelations (PFI[2020] increases from 0 to 100 as press freedom decreases, *i.e.* it could be better described as a lack-of-press-freedom parameter), there does appear to be a tendency for the countries with the lowest clustering values to have intermediate PFI[2020] $\sim 40$. In other words, despite the overall relation, some countries with the lowest levels of press freedom appear to have noise in their daily SARS-CoV-2 counts that appears only moderately low or typical.

Mainland China stands out as an exception in all eight panels of these four figures, with both a high clustering, $\phi_i = 80.35$ in the full sequence case, and a high lack of press freedom, $\text{PFI}^{2020} = 78.48$.

While a causal relation, *via* general processes of media freedom pressuring politicians and public servants to produce honest data, and vice versa, would provide the simplest interpretation of the overall correlation found here, other interpretations should be considered. Indices to measure the much wider concept of democracy tend to suffer from a lack of clarity in definitions and method (*Munck & Verkuilen, 2002*), quite likely due to the nature of democracy as a highly complex phenomenon that is difficult to represent with a single index. Nevertheless, *Balashov, Yan & Zhu (2021)* study the relations between democracy indicators and validity in daily COVID-19 data, using a very different method to the one introduced in this paper, and point out that democracy, economic and health system national indicators tend to correlate strongly to one another (see §2 of *Balashov, Yan & Zhu, 2021* for a literature review of relations between democracy and data manipulation). An alternative interpretation to direct causality could be explored along these lines. Other lines of analysis would be needed to establish causal relations instead of statistical correlations.

## CONCLUSION

Given the overdispersed, one-parameter Poissonian $\phi_i$ model proposed, the noise characteristics of the daily SARS-CoV-2 infection data suggest that most of the countries' data form a single family in the $(\phi_i, N_i)$ plane. The clustering—whether epidemiological in origin, or caused by testing or administrative pipelines—tends to be greater for greater numbers of total infections. Several countries appear, however, to show unusually anti-clustered (low-noise) daily infection counts.

Since these daily infection counts data constitute data of high epidemiological interest, the statistical characteristics presented here and the general method could be used as the basis for further investigation into the data of countries showing exceptional characteristics. The relations between the most anti-clustered counts and the psychologically significant decimal system round numbers (India: 10,000 daily, Belarus: 1,000 daily, Albania, Poland: 300 daily), and in relation to a *de facto* national presidential election, raise the question of whether or not these are just coincidences. A statistically significant anticorrelation of the clustering with the *Reporters sans frontières* Press Freedom Index was found, *i.e.*, less press freedom was found to correlate with less clustering, strengthening the credibility of the $\phi_i$ clustering model for judging the validity of daily pandemic data published by national government agencies. The suspicious periods of data found here are mostly complementary to those studied by Balashov et al., since those authors' Benford's law analysis mainly focuses on the first-digit Benford's law during the exponentially growing phases of the pandemic in any particular country (*Balashov, Yan & Zhu, 2021*), while this analysis studies noise in data for the full pandemic up to 6 May 2021.

It should be straightforward for any reader to extend the analysis in this paper by first checking its reproducibility with the free-licensed source package provided using the

Maneage framework (*Akhlaghi et al., 2021*), and then extending, updating or modifying it in other appropriate ways; see §Code availability below. Reuse of the data should be straightforward using the files archived at zenodo.4765705.

## APPENDIX A

## JHU CSSE DATA

The John Hopkins University Center for Systems Science and Engineering global time series data was downloaded on 6 May 2021 from https://raw.githubusercontent.com/CSSEGISandData/COVID-19/master/csse_covid_19_data/csse_covid_19_time_series/time_series_covid19_confirmed_global.csv, from Git commit 51CB3EE, and analysed using the same software and parameters as for the C19CCTF data. Tables A1–A4 show the equivalent of Tables 1–4, respectively. The rankings and $\phi_i$ estimates appear mostly similar between the two datasets, with small differences. One difference is that the low $\phi_i^7$ value for India shown in Table 4 is absent in Table A4. In other words, while the media stated that the daily confirmed count in India first went above the 10,000-per-day psychological threshold on 12 June 2020 (*Porecha, 2020*), the JHU CSSE data crossed this threshold earlier, and contains noise that was unknown at that time to the national Indian media and is absent from the C19CCTF data.

Another difference is that Saudi Arabia, Iran, and the United Arab Emirates have lowest-noise subsequence dates detected in 2021 in the JHU CSSE Tables A2–A4, while no country has lowest-noise subsequences in 2021 in the C19CCTF data (Tables 2–4). The relative strengths of the AIC and BIC evidence in Table A5 are similar to those in Table 5, even though the values change.

Table A6 shows that the JHU CSSE data generally find somewhat stronger anticorrelations between the clustering parameters and PFI$^{2020}$ compared to Table 6.

## SOFTWARE ACKNOWLEDGEMENTS

This research was partly done using the following free-licensed software packages: Boost 1.73.0, Bzip2 1.0.6, cURL 7.71.1, Dash 0.5.10.2, Discoteq flock 0.2.3, Eigen 3.3.7, Expat 2.2.9, File 5.39, Fontconfig 2.13.1, FreeType 2.10.2, Git 2.28.0, GNU Autoconf 2.69.200-babc, GNU Automake 1.16.2, GNU AWK 5.1.0, GNU Bash 5.0.18, GNU Binutils 2.35, GNU Compiler Collection (GCC) 10.2.0, GNU Coreutils 8.32, GNU Diffutils 3.7, GNU Findutils 4.7.0, GNU gettext 0.21, GNU gperf 3.1, GNU Grep 3.4, GNU Gzip 1.10, GNU Integer Set Library 0.18, GNU libiconv 1.16, GNU Libtool 2.4.6, GNU libunistring 0.9.10, GNU M4 1.4.18-patched, GNU Make 4.3, GNU Multiple Precision Arithmetic Library 6.2.0, GNU Multiple Precision Complex library, GNU Multiple Precision Floating-Point Reliably 4.0.2, GNU Nano 5.2, GNU NCURSES 6.2, GNU Patch 2.7.6, GNU Readline 8.0, GNU Sed 4.8, GNU Tar 1.32, GNU Texinfo 6.7, GNU Wget 1.20.3, GNU Which 2.21, GPL Ghostscript 9.52, ImageMagick 7.0.8-67, Less 563, Libbsd 0.10.0, Libffi 3.2.1, libICE 1.0.10, Libidn 1.36, Libjpeg v9b, Libpaper 1.1.28, Libpng 1.6.37, libpthread-stubs (Xorg) 0.4, libSM 1.2.3, Libtiff 4.0.10, libXau (Xorg) 1.0.9, libxcb (Xorg) 1.14, libXdmcp (Xorg) 1.1.3, libXext 1.3.4, Libxml2 2.9.9, libXt 1.2.0, Lzip 1.22-rc2, Metastore (forked) 1.1.2-

**Table A1** As in Table 1, for the JHU CSSE data: clustering parameters for the countries with the 10 lowest $\phi_i$ and 10 lowest $\psi_i$ values, *i.e.*, the least noise; extended version of table: zenodo.4765705/phi_N_full_jhu.dat

| Country | $\phi_i'$ Model | | | | | Alternative analyses | | | |
|---|---|---|---|---|---|---|---|---|---|
| | | | | | | $\widehat{v}_i$ | | $\omega_i$ | |
| | $N_i$ | $P_i^{\text{Poiss}}$ | $P_i^{\text{KS}}$ | $\phi_i$ | $\psi_i$ | $P_i^{\text{KS}}$ | $\phi_i$ | $P_i^{\text{KS}}$ | $\omega_i$ |
| Syria | 23,121 | 0.48 | 0.94 | 0.72 | 0.004 | 0.94 | 0.72 | 0.48 | 0.00 |
| Algeria | 123,272 | 0.04 | 0.19 | 0.98 | 0.002 | 0.20 | 1.00 | 0.04 | 0.00 |
| Croatia | 339,412 | 0.27 | 0.89 | 3.24 | 0.005 | 0.89 | 3.24 | 0.70 | 1.02 |
| Saudi Arabia | 422,316 | 0.00 | 0.83 | 3.67 | 0.005 | 0.66 | 3.55 | 0.62 | 2.43 |
| New Zealand | 2,637 | 0.10 | 0.88 | 3.85 | 0.074 | 0.89 | 4.68 | 0.90 | 3.63 |
| Albania | 131,419 | 0.00 | 0.16 | 4.90 | 0.013 | 0.17 | 4.90 | 0.09 | 3.76 |
| Thailand | 74,921 | 0.29 | 0.99 | 5.37 | 0.019 | 0.99 | 5.37 | 0.96 | 3.80 |
| Denmark | 257,182 | 0.00 | 0.97 | 5.56 | 0.010 | 0.99 | 5.56 | 0.91 | 5.50 |
| Iceland | 6,498 | 0.33 | 1.00 | 5.96 | 0.073 | 0.99 | 5.96 | 0.95 | 4.27 |
| Greece | 352,027 | 0.03 | 0.98 | 6.53 | 0.011 | 0.92 | 5.43 | 0.67 | 5.50 |
| Algeria | 123,272 | 0.04 | 0.19 | 0.98 | 0.002 | 0.20 | 1.00 | 0.04 | 0.00 |
| Russia | 4,792,354 | 0.00 | 0.31 | 10.12 | 0.004 | 0.26 | 9.44 | 0.26 | 8.81 |
| Syria | 23,121 | 0.48 | 0.94 | 0.72 | 0.004 | 0.94 | 0.72 | 0.48 | 0.00 |
| Croatia | 339,412 | 0.27 | 0.89 | 3.24 | 0.005 | 0.89 | 3.24 | 0.70 | 1.02 |
| Saudi Arabia | 422,316 | 0.00 | 0.83 | 3.67 | 0.005 | 0.66 | 3.55 | 0.62 | 2.43 |
| Iran | 2,591,609 | 0.00 | 0.33 | 11.61 | 0.007 | 0.17 | 10.00 | 0.25 | 9.66 |
| Turkey | 4,955,594 | 0.00 | 0.02 | 19.95 | 0.008 | 0.01 | 19.27 | 0.01 | 16.98 |
| Denmark | 257,182 | 0.00 | 0.97 | 5.56 | 0.010 | 0.99 | 5.56 | 0.91 | 5.50 |
| Hungary | 785,967 | 0.02 | 0.99 | 9.23 | 0.010 | 0.98 | 14.29 | 0.91 | 7.00 |
| Belarus | 363,732 | 0.00 | 0.01 | 6.92 | 0.011 | 0.01 | 6.46 | 0.01 | 5.13 |

23-fa9170b, OpenBLAS 0.3.10, Open MPI 4.0.4, OpenSSL 1.1.1a, PatchELF 0.10, Perl 5.32.0, pkg-config 0.29.2, Python 3.8.5, Unzip 6.0, util-Linux 2.35, util-macros (Xorg) 1.19.2, X11 library 1.6.9, XCB-proto (Xorg) 1.14, xorgproto 2020.1, xtrans (Xorg) 1.4.0, XZ Utils 5.2.5, Zip 3.0 and Zlib 1.2.11. Python packages used include: Cycler 0.10.0, Cython 0.29.21 (*Behnel et al., 2011*), Kiwisolver 1.0.1, Matplotlib 3.3.0 (*Hunter, 2007*), Numpy 1.19.1 (*Van der Walt, Colbert & Varoquaux, 2011*), pybind11 2.5.0, PyParsing 2.3.1, python-dateutil 2.8.0, Scipy 1.5.2 (*Oliphant, 2007*; *Millman & Aivazis, 2011*), Setuptools 41.6.0, Setuptools-scm 3.3.3 and Six 1.12.0. LaTeX packages for creating the pdf version of the paper included: algorithmicx 15878 (revision), algorithms 0.1, biber 2.16, biblatex 3.16, bitset 1.3, booktabs 1.61803398, breakurl 1.40, caption 56771 (revision), changepage 1.0c, courier 35058 (revision), csquotes 5.2l, datetime 2.60, dblfloatfix 1.0a, ec 1.0, enumitem 3.9, epstopdf 2.28, eso-pic 3.0a, etoolbox 2.5k, fancyhdr 4.0.1, float 1.3d, fmtcount 3.07, fontaxes 1.0e, footmisc 5.5b, fp 2.1d, kastrup 15878 (revision), lastpage 1.2m, latexpand 1.6, letltxmacro 1.6, lineno 4.41, listings 1.8d, logreq 1.0, microtype 2.8c, multirow 2.8, mweights 53520 (revision), newtx 1.642, pdfescape 1.15, pdftexcmds 0.33, pgf 3.1.8b, pgfplots 1.17, preprint 2011, setspace 6.7a, soul 2.4, sttools 2.1, subfig 1.3, tex

**Table A2** As in Table 2, for the JHU CSSE data: least noisy 28-day sequences—clustering parameters for the countries with the 10 lowest $\phi_i$ values; extended table: zenodo.4765705/phi_N_28days_jhu.dat.

| Country | $N_i$ | $\psi_i$ | $P_i^{\text{Poiss}}$ | $P_i^{\text{KS}}$ | $\phi_i^{28}$ | Starting date |
|---|---|---|---|---|---|---|
| Algeria | 123,272 | 338.2 | 0.02 | 0.72 | 0.05 | 2020-08-18 |
| Turkey | 4,955,594 | 1,014.5 | 0.03 | 1.00 | 0.14 | 2020-06-30 |
| United Arab Emirates | 529,220 | 2,884.9 | 0.01 | 0.07 | 0.15 | 2020-12-30 |
| Belarus | 363,732 | 921.9 | 0.14 | 0.89 | 0.21 | 2020-05-08 |
| Albania | 131,419 | 203.8 | 0.33 | 0.64 | 0.23 | 2020-09-27 |
| Russia | 4,792,354 | 5,414.0 | 0.36 | 0.85 | 0.24 | 2020-07-19 |
| Saudi Arabia | 422,316 | 332.5 | 0.54 | 0.78 | 0.43 | 2021-02-01 |
| Syria | 23,121 | 70.0 | 0.19 | 0.91 | 0.50 | 2020-08-15 |
| Iran | 2,591,609 | 6,594.5 | 0.14 | 0.41 | 1.51 | 2021-01-15 |
| Georgia | 315,913 | 384.4 | 0.79 | 0.99 | 1.66 | 2020-09-17 |

**Table A3** As in Table 3, for the JHU CSSE data: least noisy 14-day sequences—clustering parameters for the countries with the 10 lowest $\phi_i^{14}$ values; extended version of table: zenodo.4765705/phi_N_14days_jhu.dat.

| Country | $N_i$ | $\langle n_i^{14} \rangle$ | $P_i^{\text{Poiss}}$ | $P_i^{\text{KS}}$ | $\phi_i^{14}$ | Starting date |
|---|---|---|---|---|---|---|
| United Arab Emirates | 529,220 | 3384.1 | 0.07 | 0.35 | 0.05 | 2021-01-11 |
| Algeria | 123,272 | 336.4 | 0.06 | 0.80 | 0.05 | 2020-08-26 |
| Turkey | 4,955,594 | 971.6 | 0.12 | 0.86 | 0.11 | 2020-07-08 |
| Belarus | 363,732 | 945.6 | 0.22 | 1.00 | 0.13 | 2020-05-12 |
| Albania | 131,419 | 143.4 | 0.16 | 0.92 | 0.15 | 2020-09-01 |
| Saudi Arabia | 422,316 | 337.7 | 0.32 | 0.79 | 0.20 | 2021-02-08 |
| Russia | 4,792,354 | 5165.5 | 0.47 | 0.51 | 0.28 | 2020-08-01 |
| Syria | 23,121 | 76.6 | 0.42 | 0.96 | 0.35 | 2020-08-14 |
| Poland | 2,811,951 | 299.9 | 0.55 | 0.68 | 0.53 | 2020-06-17 |
| Kenya | 161,393 | 126.2 | 0.54 | 0.91 | 0.57 | 2020-06-03 |

3.141592653, texgyre 2.501, times 35058 (revision), titlesec 2.13, trimspaces 1.1, txfonts 15878 (revision), ulem 53365 (revision), varwidth 0.92, wrapfig 3.6, xcolor 2.12, xkeyval 2.8 and xstring 1.83.

## DATA AVAILABILITY

As described above in "SARS-CoV-2 Infection Data", the source of curated SARS-CoV-2 infection count data used for the main analysis in this paper is the C19CCTF data, downloaded using the script download-wikipedia-SARS-CoV-2-charts.sh and stored in the file Wikipedia_SARSCoV2_charts.dat in the reproducibility package available at zenodo.4765705. The data file is archived at zenodo.4765705 /WP C19CCTF SARSCoV2. The WHO data that was compared with the C19CCTF data *via* a jump analysis (Fig. 1) was downloaded from https://covid19.who.int/WHO-COVID-19-global-data.csv and archived on 6 May 2021 at https://web.archive.org/web/20210506113321/https://covid19.who.int/WHO-COVID-19-global-data.csv.

**Table A4** As for **Table 4**, for the JHU CSSE data: least noisy 7-day sequences—clustering parameters for the countries with the 10 lowest $\phi_i^{14}$ values; extended table: **zenodo.4765705/phi_N_07days_jhu.dat**.

| Country | $N_i$ | $\langle n_i^{14} \rangle$ | $P^{Poiss}_i$ | $P^{KS}_i$ | $\phi_i^7$ | Starting date |
|---|---|---|---|---|---|---|
| United Arab Emirates | 529,220 | 544.9 | 0.24 | 0.99 | 0.05 | 2020-04-27 |
| Turkey | 4,955,594 | 929.6 | 0.22 | 0.93 | 0.05 | 2020-07-15 |
| Albania | 131,419 | 297.7 | 0.23 | 0.98 | 0.05 | 2020-10-18 |
| Belarus | 363,732 | 947.9 | 0.60 | 0.94 | 0.05 | 2020-05-13 |
| Algeria | 123,272 | 204.3 | 0.37 | 0.49 | 0.05 | 2020-10-14 |
| Russia | 4,792,354 | 5,035.0 | 0.38 | 0.75 | 0.10 | 2020-08-09 |
| Poland | 2,811,951 | 297.0 | 0.51 | 0.99 | 0.10 | 2020-06-20 |
| Saudi Arabia | 422,316 | 175.6 | 0.52 | 0.99 | 0.15 | 2021-01-13 |
| Syria | 23,121 | 82.3 | 0.21 | 0.97 | 0.17 | 2020-08-14 |
| Panama | 365,975 | 171.1 | 0.82 | 0.96 | 0.17 | 2020-05-09 |

**Table A5** As for **Table 5**, *Akaike (1974)* and Bayesian (*Schwarz, 1978*) information criteria for the $\phi_i'$ and alternative analyses for the JHU CSSE data; plain-text version: **zenodo.4765705/AIC_BIC_full_jhu.dat**.

| Model | $\phi_i'$ | | Log. median | | Neg. binomial | | 2-Day grouping | | 3-Day grouping | |
|---|---|---|---|---|---|---|---|---|---|---|
| | AIC | BIC | AIC | BIC | AIC | BIC | AIC | BIC | AIC | BIC |
| | 376.18 | 994.94 | 401.69 | 1,020.44 | 498.00 | 1116.75 | 421.96 | 1,032.94 | 239.83 | 811.89 |

**Table A6** As for **Table 6**, Kendall $\tau$ statistic and its significance (two-sided) $P^\tau$ for the null hypothesis of no correlation between the ranking of PFI$^{2020}$ and $\phi_i$ or $\psi_i$ for the full data or subsequences, for the JHU CSSE data; plain-text version: **zenodo.4765705/pfi_correlations_table_jhu.dat**.

| Parameter | Full | | 28-day | | 14-day | | 7-day | |
|---|---|---|---|---|---|---|---|---|
| | $\tau$ | $P^\tau$ | $\tau$ | $P^\tau$ | $\tau$ | $P^\tau$ | $\tau$ | $P^\tau$ |
| $\phi_i$ | −0.124 | 0.105 | −0.158 | 0.0400 | −0.175 | 0.0230 | −0.232 | 0.00254 |
| $\psi_i$ | −0.165 | 0.0318 | −0.162 | 0.0346 | −0.163 | 0.0339 | −0.194 | 0.0112 |

## CODE AVAILABILITY

In addition to the SARS-CoV-2 infection count data for this paper, the full downloading of complementary data, calculations, production of figures, tables and values quoted in the text of the pdf version of the paper are intended to be fully reproducible on any POSIX-compatible system using free-licensed software, which, by definition, the user may modify, redistribute, and redistribute in modified form. The reproducibility framework is technically a git branch of the Maneage package (*Akhlaghi et al., 2021*) (https://maneage.org), earlier used to produce reproducible papers (*Infante-Sainz, Trujillo & Roman, 2020*). The git repository commit ID of this version of this paper is subpoisson-f72cb84. The primary (live) git repository is https://codeberg.org/boud/subpoisson, archived at

swh:1:rev:789e651c0fb23b2585555c08de1b44d9e25cfb6d. The full reproducibility package is archived at zenodo.4765705. Bug reports and discussion are welcome at https://codeberg.org/boud/subpoisson/issues.

# ACKNOWLEDGEMENTS

Thank you to Marius Peper, Taha Rouabah, Dmitry Borodaenko, anonymous colleagues, and to Niayesh Afshordi and the two other referees for several useful comments and to the Maneage developers for the Maneage framework in general and for several specific comments on this work.

## Funding

This project was supported with computational resources by the Poznan Supercomputing and Networking Center (PSNC) computational grant 314. The PSNC had no role in study design, data collection and analysis, decision to publish, or preparation of the manuscript.

## Grant Disclosures

The following grant information was disclosed by the authors:
Poznan Supercomputing and Networking Center (PSNC): 314.

## Competing Interests

The author declares that he has no competing interests.

## Author Contributions

- Boudewijn F. Roukema conceived and designed the experiments, performed the experiments, analyzed the data, prepared figures and/or tables, authored or reviewed drafts of the paper, and approved the final draft.

## Data Availability

The source of curated SARS-CoV-2 infection count data used for the main analysis in this article is the C19CCTF data, available using the script download-wikipedia-SARS-CoV-2-charts.sh and stored in the file Wikipedia SARSCoV2 charts.dat in the reproducibility package available at https://zenodo.org/record/4765705.

The data file is archived at https://zenodo.org/record/4765705/files/WP_C19CCTF_SARSCoV2.dat.

The WHO data that was compared with the C19CCTF data *via* a jump analysis (Fig. 1) was downloaded from https://covid19.who.int/WHO-COVID-19-global-data.csv and archived on 6 May 2021 at https://web.archive.org/web/20210506113321/https://covid19.who.int/WHO-COVID-19-global-data.csv.

The complementary data, calculations, production of figures, tables and values are intended to be fully reproducible on any POSIX-compatible system using free-licensed software.

The reproducibility framework is technically a GIT branch of the MANEAGE package (https://maneage.org; *Akhlaghi et al., 2021*), earlier used to produce reproducible papers (*Infante-Sainz, Trujillo & Roman, 2020*).

The GIT repository commit ID of this version of this article is subpoisson-f72cb84. The primary (live) GIT repository (https://codeberg.org/boud/subpoisson) is archived at https://archive.softwareheritage.org/browse/revision/789e651c0fb23b2585555c08de1b44d9e25cfb6d.

The full reproducibility package is archived at Zenodo: Roukema, Boudewijn F. (2021). Anti-clustering in the national SARS-CoV-2 daily infection counts (Version 72242ca). Peerj, in press. DOI 10.5281/zenodo.4765705.

Bug reports and discussion are welcome at https://codeberg.org/boud/subpoisson/issues.

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
