# Peer review of "Anti-clustering in the national SARS-CoV-2 daily infection counts"

_PeerJ, doi:10.7717/peerj.11856_

## Round 0.1 · original submission · Major Revisions

There are several issues highlighted by the reviewers, that you could address in a revised version of the text.

Reviewer 1 ·

Basic reporting

OK

Experimental design

See comments on analysis method below

Validity of the findings

See comment below

Additional comments

In this manuscript, the authors analyze data regarding the national daily COVID-19 infection number from the Wikipedia WikiProject COVID-19 Case Count Task Force data. They look for a criteria to establish if the noise in the data is consistent with reliably reported data. As a bench mark, they postulate that the noise should correspond to Poissonian noise, and test it by a one-parameter Poissonian \phi_i model, which basically assumes that the number of daily infections should grouped in clusters, each containing\phi_i counts per cluster. \phi_i is determined in different regions by choosing the value which leads to the best fit to Poissonian distribution. Applying this analysis to different countries reveals that for most countries a super-Poissonian behavior emerges.

Since much of the decisions relating to COVID-19 policies are based on daily infection numbers, it is worthwhile to develop a better understanding of their statistical characteristics, which can be used to draw attention to questionable reporting or unusual circumstances. Therefore, there is no doubt that the manuscript is timely and of interest.

Nevertheless, I feel less confident about the assumptions and method. Based on my experience of analyzing noisy data from complex dynamical systems (albeit very different than the one discussed here), Poisson noise usually arise in non-correlated simple dynamics. For correlated complex systems (which I suspect corresponds also to the infection dynamics) I expect different noise characteristics. Indeed this seems to be the authors conclusion from their analysis. What I am missing is why did they base their analysis to begin with on a deviation from Poisson noise .Moreover, once it is clear that Poisson noise does not capture the behavior, why use a one-parameter Poissonian \phi_i model, where \phi_i is a single cluster size, although one would expect a wide range of clustering (power law?) to describe the data?

I believe that the authors should clarify their rational for choosing their analysis method more clearly, and if possible, supplement them by an analysis more suitable for super Poissonian (clustered) noise. This will greatly enhance the value of their study.

·

Basic reporting

This is a very interesting analysis, focusing on the properties of noise in epidemic reporting, which is an often under-appreciated source of information. While it does provide a basic notion super-Poissonian vs sub-Poissonian noise, it appears to does side-step several issues that may warrant further discussion. I will comment on these below.

Experimental design

1. The assumption that the expected value of cases on one day is the median of 4 neighboring days can lead to significant overestimate during exponential growth or decay. I suggest fitting an exponential (or a quadratic form for log(n_j)) instead, to test for this effect.
2. Statement in Line 130, suggesting that relative noise in Poisson process is 1/\sqrt{N} is only for large N, and the exact Poisson probability \lambda^N\exp(-\lambda)/N!, can be used for any N, including N=0.
3. I do not understand why a KS test is needed to determine \phi'. A product of (rescaled) Poisson probabilities gives the likelihood for \phi', which can be used to constrain \phi' using Bayes theorem.
4. What does "sufficient fit" in Line 201 mean?
5. Many reported cases have weekly cycles (e.g. a lull in weekends). Is there a way to test for this effect?
6. Does the analysis apply to multiple epidemic waves?
7. While the article focuses on daily reporting, it would be useful to see how the results change if we look at longer periods, such as those consisting of multiple days
8. Translation of acronyms to country names are missing from the manuscript
9. Link in Line 429 is broken

Validity of the findings

The main premise of this paper is that some countries appear as outliers in the distributions of a statistical measure, and they are interpreted as due to (intentional or unintentional) administrative reporting practices. What is missing from this is any measure of the epidemic itself, which has proved highly unpredictable and chaotic, and can vary from region to region, for a multitude of reasons. This seems premature. For example, the US and Brazil exceeding the extrapolation from small epidemics may well be a feature of a larger more heterogenous epidemic, leading to more daily variations. The grouping of low variance vs high variance countries appears arbitrary and does not follow from a clear quantitative criterion.

Line 320 says "It is in this sense that the sequence can be considered sub-Poissonian. ", through comparison of Algeria with other countries, even though the data is NOT actually sub-Poissonian. As such, I believe this terminology is misleading: sub-Poissonian suggest doctored data, while Poissonian noise may be natural for the epidemic in one country, even though it may not be in others. The COVID-19 pandemic has shown a huge diversity of behavior across different countries and continents, and simply being an outlier (e.g. New Zealand) doesn't suggest doctored data.

Overall, my recommendation is discuss the extent to which the underlying unknown epidemic dynamics may confound the findings here. Furthermore, I recommend a quantitative p-value, with a clear threshold, that could be used to reject Poissonian and super-Poissonian hypotheses.

Reviewer 3 ·

Basic reporting

The paper is well written and the flow of the paper is logical.

The paper lacks comparison with the vast non parametric statistics literature that can be used to assess the proposed anti-clustering behavior.

Experimental design

The research question is well defined. The proposed solution is statistically weak, for at least three reasons:

1. One of the many non parametric tests available should be attempted (see any introductory statistics book that have. chapter on non parametric statistic)

2. Taking the median of the counts over five consecutive days to estimate the Poisson parameter is data-driven and ad hoc assumption which may bias the results of the Kolmogorov test. The Authors should avoid this and use a time dependent model such as an autoregressive one

3. The issue of over/under dispersion wrt to the Poisson can be tackled with a negative binomial distribution, and several authors have used this to model covid-19 count data

Validity of the findings

The findings are well described. However, the methodology on which they are based is weak and there is no comparison with alternative non parametric or semi parametric methods

Additional comments

The problem of anti-clustering of the covid-19 data is a relevant one. The methods proposed to ascertain statistically the presence of an anti-clustering behavior in covid-19 national data of a set of countries is weak and there is no comparison with alternative methods. The background statistical literature is very scarce. The paper should be rejected in the present form.

---

## Round 0.2 · Minor Revisions

Some minor changes are suggested by one of the reviewers.

Reviewer 3 ·

Basic reporting

The paper concerns a relevant problem, it is original and well written. The authors have taken my previous comments into account in their revised version

Experimental design

The experimental design is clear and improved after the revision

Validity of the findings

After the revision the findings are more robust

Additional comments

The paper could be accepted conditionally on a further enlargement of the bibliographic references, especially on the comments introduced after revision

For example, concerning autoregressive methods, the authors should quote Agosto,Campmas, Giudici, Renda,
Monitoring covid-19 contagion growth, statistics in medicine, 2021:
https://doi.org/10.1002/sim.9020

And the references therein.

---

## Round 0.3 · accepted · Accept

All the reviewers' concerns have been correctly addressed.

Reviewer 3 ·

Basic reporting

no comment

Experimental design

no comment

Validity of the findings

no comment

Additional comments

The Authors have satisfactorily addressed my previous comments.